# Isogenic Cell Lines Derived from Specific Organ Metastases Exhibit Divergent Cytogenomic Aberrations

**DOI:** 10.3390/cancers15051420

**Published:** 2023-02-23

**Authors:** Paul T. Winnard, Laura Morsberger, Raluca Yonescu, Liqun Jiang, Ying S. Zou, Venu Raman

**Affiliations:** 1Department of Radiology and Radiological Sciences, Johns Hopkins University School of Medicine, Baltimore, MD 21205, USA; 2Department of Pathology, Johns Hopkins University School of Medicine, Baltimore, MD 21205, USA; 3Department of Oncology, Johns Hopkins University School of Medicine, Baltimore, MD 21205, USA; 4Pharmacology and Molecular Sciences, Johns Hopkins University School of Medicine, Baltimore, MD 21205, USA; 5Department of Pathology, University Medical Center Utrecht, 3584 CX Utrecht, The Netherlands

**Keywords:** karyotypes, isogenic metastatic cell lines, inter- and intra-cytogenomic heterogeneity, comparisons

## Abstract

**Simple Summary:**

Normal human cells have 22 pairs of chromosomes as well as 2 sex chromosomes for a total of 46 chromosomes; this normal karyotype is called diploidy (euploidy). On the other hand, aberrant numbers of chromosomes, i.e., gains and/or losses of chromosomes, have been found in most human cancer cells. This condition is called aneuploidy. Within in a clinical context, aneuploidy has been shown to be a marker of poor prognosis and drug resistance. Importantly, the deadliest stage of a cancer occurs when the cancer has been found to have spread from a primary tumor site to other organ sites, which is called metastasis. Controlled comprehensive clinical studies of metastatic cancer, which require an interrogation of the affected site(s), such as lungs, or liver, brain, or bone, with the goal of developing better treatment are very challenging. Therefore, repeatable controlled studies of complex human metastatic disease are simulated in animal systems using human cancer cells in special mouse strains. We used such a model system to better understand the chromosomal changes and the processes that bring them about, along with a study of gene variants, chromosomal amplifications, gains, and losses in metastatic cancer cells. We compared these differences to their primary tumor cell counterparts. This information aids us in suggesting possible new therapeutic treatments that may have a potential to limit the growth of metastatic cancer.

**Abstract:**

Aneuploidy, a deviation in chromosome numbers from the normal diploid set, is now recognized as a fundamental characteristic of all cancer types and is found in 70–90% of all solid tumors. The majority of aneuploidies are generated by chromosomal instability (CIN). CIN/aneuploidy is an independent prognostic marker of cancer survival and is a cause of drug resistance. Hence, ongoing research has been directed towards the development of therapeutics aimed at targeting CIN/aneuploidy. However, there are relatively limited reports on the evolution of CIN/aneuploidies within or across metastatic lesions. In this work, we built on our previous studies using a human xenograft model system of metastatic disease in mice that is based on isogenic cell lines derived from the primary tumor and specific metastatic organs (brain, liver, lung, and spine). As such, these studies were aimed at exploring distinctions and commonalities between the karyotypes; biological processes that have been implicated in CIN; single-nucleotide polymorphisms (SNPs); losses, gains, and amplifications of chromosomal regions; and gene mutation variants across these cell lines. Substantial amounts of inter- and intra-heterogeneity were found across karyotypes, along with distinctions between SNP frequencies across each chromosome of each metastatic cell line relative the primary tumor cell line. There were disconnects between chromosomal gains or amplifications and protein levels of the genes in those regions. However, commonalities across all cell lines provide opportunities to select biological processes as druggable targets that could have efficacy against the primary tumor, as well as metastases.

## 1. Introduction

Aneuploidy, a deviation in chromosome numbers from the normal diploid set, has a long history and was first described 130 years ago from observations in fresh human carcinoma specimens [1,2]. It is now recognized as a fundamental characteristic of all cancer types and is found in 70–90% of all solid tumors [3,4,5]. Consequently, cancer genomes exhibit massive aberrations in copy number changes due to losses or gains in whole chromosomes or chromosome arms that result in numerical and structural chromosomal changes. As such, aneuploidy reflects extensive genetic defects that exceed levels of any other genetic lesion [6]. The majority of aneuploidies are generated by chromosomal instability (CIN), which has been found to be generated by a variety of mechanisms [7,8,9,10,11,12,13,14,15,16,17,18,19,20,21,22,23]. However, it has been noted that aneuploidy can arise independent of CIN [6]. Importantly, it has been repeatedly demonstrated that CIN/aneuploidy is an independent prognostic marker of cancer survival [4,24,25,26,27] and is a cause of drug resistance [28,29,30]. Hence, ongoing research has been directed towards the development of therapeutics aimed at targeting CIN/aneuploidy as a means of overcoming chemotherapy resistance and prolonging survival [6,29,31,32,33,34,35,36]. It is important to note that the bulk of research has been focused on primary tumor samples or their cell lines. Hence, there are relatively limited reports on the evolution of CIN/aneuploidies within metastatic lesions and how the resulting aneuploidies compare to the aneuploidies of their primary tumors [9,37,38,39], which has left gaps in our knowledge, particularly from the perspective of alternative treatment strategies for metastatic disease.

In this work, we built on our previous omics studies on a human xenograft model system of metastatic disease in mice [40,41,42]. This model system generated isogenic cell lines derived from the primary tumor and specific metastatic organs (brain, liver, lung, and spine), which enabled a comparison of proteomes, transcriptomes, and metabolomes, as well as associated pathways across all isogenic cell lines. Those studies revealed commonalities, along with important tissue-specific divergencies in protein, mRNA, metabolites, pathways, and drug sensitivities [40]. The studies reported here were aimed at exploring distinctions and commonalities between the karyotypes; biological processes that have been implicated in CIN/aneuploidy; losses, gains, and amplifications of chromosomal regions, i.e., further indications of CIN; single-nucleotide polymorphisms (SNPs); and gene mutation variants that may reflect gene-level instabilities across these cell lines. Substantial amounts of inter- and intra-heterogeneity were found across karyotypes, along with distinctions between SNP frequencies across each chromosome of each metastatic cell line relative to the primary tumor cell line. There were disconnects between chromosomal gains or amplifications and protein levels of the genes in those regions. Overall, our analyses underscore the complexity of tissue-specific differential distinctions between all cell lines from the level of the genome (i.e., aberrant karyotypes) and gene (differences in SNP signatures and mutant variants) to transcript- and protein-level differences within the context of biological processes, which, if dysregulated, mediate CIN. However, commonalities across all cell lines provide opportunities to select biological processes or gains and amplifications as druggable targets that could have efficacy against the primary tumor, as well as metastases. 

## 2. Methods

### 2.1. Cell Lines

Generation and characterization of the parental MDA-MB-435-tdTomato (435-tdT) fluorescent cell line and subsequent isogenic primary (1°) tumor and metastatic cell lines have been previously described [41,43]. Briefly, orthotopic 1° tumor xenografts were initiated by injection of 435-tdT (2 × 10^6^) cells into the second thoracic mammary fat pad of 5 female NOD-SCID mice. After 13–15 weeks of tumor growth, the mice were sacrificed and, 1° tumor, brain, liver, lungs, and spine were immediately excised, dissected away from fat and muscle, and placed into sterile phosphate-buffered saline on ice. All organs/bones were inspected using fluorescence microscopy for any signs of metastatic burden, which was easily discerned as bright tdT red fluorescence. Areas of fluorescence, along with adjacent tissue, were cut away and placed into 100 mm cell culture plates in 10 mL sterile medium and then immediately minced within a sterile hood. 

All tissue explants were initially cultured in Roswell Park Memorial Institute (RPMI)-10% fetal bovine serum (FBS) medium supplemented with antibiotics (100 I.U./mL penicillin (pen), 100 μg/mL streptomycin (strep), 100 μg/mL ampicillin, and 100 μg/mL kanamycin) and, as necessary, Fungizone. Medium was refreshed every 2–3 days, and after 2 weeks of culture, the medium was changed to RPMI-10% FBS supplemented with pen/strep. Further studies resulted in optimal media selections: Dulbecco’s modified Eagle medium (DMEM-10% FBS) for the parental cell line and DMEM:Ham’s F12 (50:50)-5% FBS for the 1° tumor and all metastatic cell lines. Cells were cultured in standard humidified incubators at 37 °C and 5% CO_2_.

### 2.2. Proteomics, RNA-Seq, and Exome-Associated SNPs

Proteomics were performed from a single sampling of each cell line’s proteins in the Mass Spectroscopy and Proteomics Facility at the Johns Hopkins University Medical School using tandem mass tags (TMTs) for direct comparisons of all 10 samples in a single tandem MS experiment, as previously described [40]. 

RNA-seq was performed from a single sampling of each cell line’s RNA at a commercial facility (BGI Americas, San Jose, CA, USA), as previously described [40]; the exome single-nucleotide polymorphism (SNP) datasets from 2 biological replicates were part of the RNA-seq sequencing results.

### 2.3. Karyotyping

Conventional G-banded chromosome studies were performed using standard techniques. Cells in the exponential phase of growth were incubated with colchicine (Sigma-Aldrich, St. Louis, MO, USA) to a final concentration of 0.8 μg/mL for 4 h and harvested. Cells were then treated with a hypotonic solution of potassium chloride (0.075 mol/L) and incubated at 37 °C for 30 min, fixed in acetic acid:methanol (1:3, *v*:*v*), mounted on grease-free chilled 4 °C slides, and air-dried. Giemsa–trypsin banding was performed for chromosome examination. One hundred mitotic cells per sample were analyzed. The abnormal karyotypes were described using the International System for Human Cytogenomic Nomenclature (ISCN 2020). 

### 2.4. Analysis of Genes/Proteins with Validated Functions in Cell Division 

Aneuploidy and chromosomal instability (CIN) are tightly linked, with the former being generated during a loss of high-fidelity cell division, which is a function of the latter [27,44]. Therefore, to better understand and put into perspective any dysregulations of cell division processes that could participate in the generation of the patterns of differential aneuploidies observed across our isogenic cell line model system, we utilized our proteomic and RNA-seq datasets for comparison analyses of the expression levels of 469 proteins/genes from a recently complied list of 701 proteins, which were shown to function in biological processes that are necessary for passage through the synthesis (S), G2, and mitosis (M) periods of cell division [45]. Although the entire set of 701 proteins was validated with respect to functioning during S/G2-mitosis, a subset of 469 proteins was selected because they have well-characterized known functions in cell division processes, while the remaining proteins were described as more recent additions, with some being reported for the first time [45]. We compared the linear fold change (F.C.) of the expression levels of these proteins and their transcripts for the cases of the 1° tumor cell line vs. parental cell line and each of the metastatic cell lines vs. the 1° tumor cell line. We selected and catalogued proteins and transcripts across a broad F.C. range of ≤−1.25 ≥1.25 to be consistent with our previous analyses [40,41] but focused on and stressed moderate-to-high F.C.s, i.e., those ≤−1.5 (see Result Section 3.2 and Discussion). The biological processes with which these proteins were functionally associated were cell cycle regulation, centrosome regulation, cytokinesis, chromosome partition, DNA condensation, kinetochore formation, microtubule regulation, nuclear envelope regulation, spindle assembly and regulation, spindle checkpoint, DNA damage, DNA replication, DNA metabolism, and chromatin organization [45].

### 2.5. Mutation and Copy Number Variants (Amplifications/Gains and Losses) by DNA-Based Next-Generation Sequencing

The targeted next-generation sequencing (NGS) assay has been previously described [46,47]. DNA was extracted from cell lines by conventional methods (Qiacube; Qiagen, Hilden, Germany), and DNA concentration was assessed using a Qubit fluorometer (Thermo Fisher Scientific, Waltham, MA, USA). Library preparation was performed using Kapa Roche HyperPrep reagents (Roche Diagnostics, Inc., Wilmington, MA, USA); hybrid capture used 40,670 probes with provided reagents (Integrated DNA Technologies, Inc., Coralville, IA, USA), and products were sequenced using a NovaSeq 6000 with NovaSeq Rapid Cluster and SBS v2 200-cycle reagents with Illumina paired-end technology (Illumina, Inc., San Diego, CA, USA). An in-house variant and copy number variant (CNV) caller software (MDL VC 10) and CNV kit software version 0.9.6 (https://cnvkit.readthedocs.io/en/stable, last accessed 30 June 2022) were used to generate variants (single-nucleotide variants, insertion–deletion variants, etc.) and genome-wide copy number discovery from the targeted NGS data. Only specimens with more than a 1000× unique sequencing read depth were processed through gene variant/mutation and CNV analysis pipelines. Copy number variants were determined using autosomal log2 ratio thresholds set at 1.3, 0.6–1.0, and −1.0 for the detection of amplification, gain, and loss, respectively. Analysis was performed using human reference sequence genome assembly hg19 (National Center for Biotechnology Information build GRCh37/hg19).

## 3. Results

### 3.1. Inter- and Intrakaryotype Heterogeneities across All Cell Lines

Conventional cytogenomic analyses revealed complex karyotypes with multiple structural and numerical chromosome abnormalities across all cell lines. All cell lines were hyperdiploid with similar modal numbers of chromosomes of 56, 56, 56, 55, and 56 for the 1° tumor, brain, liver, lung, and spine metastatic cell lines, respectively. Nevertheless, comprehensive karyotyping analyses provided evidence of a vast amount of intra- and intercell line karyotype heterogeneities, with the overall numbers of chromosomes ranging from 54 to 58 (Figure 1, Figure 2 and Figure 3). As seen in Figure 1, the modal karyotype (outlined in red, panel Pa-1) for the parental cell line differed substantially from four (panels: Pa-2–Pa-5) representative examples of distinctly different karyotypes, i.e., intrakaryotype heterogeneity (red arrows), found in the same population of cells. Figure 1 also shows a pattern of intrakaryotype heterogeneity (blue arrows) in the 1° tumor cell line population (panels: Tu-1–Tu-5) when comparing its model karyotype (outlined in blue, panel: Tu-1) to four different representative karyotypes (panels: Tu-2–Tu-5), which indicates that all five karyotypes are different. Figure 1 also demonstrates a substantial intercell line heterogeneity between the modal parental karyotype and all five 1° tumor karyotypes (red arrows). The modal parental karyotype exhibited numerical chromosome abnormalities, such as gains of chromosomes 1, 2, 3, 4, 5, 7, 9, 11, and 15; heterozygous (one-copy) loss of chromosomes 8, 14, 19, and 21; and homozygous (two-copy) loss of chromosome 13, as well as structural chromosome abnormalities, including a derivative 1;7 chromosome; an isochromosome 7q; additions of genetic material to chromosomes 1q21, 3q12, 11p14, 15p11.2, 18p11.2, 19p13, and 20q13.2; a duplication of chromosome 6p, which leads to a net imbalance of four copies of the 6p21.3-p22 segment; a paracentric inversion of 9q; a terminal deletion of chromosome 12p; and a gain of seven unidentified marker chromosomes (M1–M7). The modal 1° tumor’s karyotype differed from the modal parental karyotype, with a homozygous loss of chromosome 8, heterozygous losses of chromosomes 6 and 22, and a gain of four unidentified marker chromosomes (M8–M11) (Figure 1). Figure 2 shows, in the upper left-hand panel, the modal karyotype of the 1° tumor cell line, which is outlined in blue and separated from the other karyotypes in the figure by a black border along its bottom and right sides. The modal karyotype of the metastatic brain cell line (panel: Br-1) to the right of the modal 1° tumor karyotype is outlined in orange.

Compared to the modal 1° tumor karyotype, the modal brain karyotype had a gain of two abnormal number 6 chromosomes characterized by additional material added to the 6q13 arm, i.e., a gain of 6q11-6q13, a loss of five marker chromosomes (M5, M7–M9, and M11, blue type), and a gain of two novel marker chromosomes (M13 and M, blue arrows). Inter- and intrakaryotype heterogeneities between the modal karyotype of the 1° tumor cell line and the five brain cell line karyotypes, as well as between the model brain cell line karyotype and four (panels: Br-2–Br-5) additional representative brain cell line karyotypes, are indicated by blue and orange arrows, respectively, and it can be noted that the five brain cell lines are distinctly different. The five right-hand-side karyotypes in Figure 2 are the modal karyotypes of the metastatic liver cell line (panel: Li-1) outlined in violet, along with four (panels: Li-2–Li-5) other liver cell line karyotypes. In comparison to the modal 1° tumor karyotype, the modal liver cell line karyotype had a heterozygous loss of chromosome 10 (blue arrow), along with four marker chromosomes (M2, M5, M9, and M10 in blue type) and gained five novel marker chromosomes (M12, M13, and Ms, blue arrows). The distinctions between the liver cell line’s modal karyotype (outlined in violet, panel: Li-1) and the four (panels: Li-2–Li-5) other liver karyotypes are indicated by violet arrows, while the interkaryotype differences between these karyotypes and the modal 1° tumor karyotype are indicated by blue arrows. The variations between the brain and liver cell lines’ karyotypes are not indicated due to the complexity of the comparisons between 10 karyotypes; however, it can be noted that none of the representative brain and liver karyotypes are the same. In Figure 3, a comparison between the modal karyotype of the 1° tumor cell line (upper-left-hand side), the modal karyotypes of the lung cell line (panel: Lu-1, outlined in green), and the modal karyotype of the spine cell line (panel: Sp-1, outlined in dark red) again illustrate the intra- and interkaryotype distinctions between these cell lines. Thus, comparison between the modal karyotype of the 1° tumor and the modal karyotype of the lung cell line (panel: Lu-1) indicates that the latter loses abnormal chromosomes 7 and 15 (blue arrows) and two marker chromosomes (M9 and M10, blue type) and gains chromosomes 4 and 6 (blue arrows), along with a novel marker chromosome (M, blue arrow). Finally, relative to the modal 1° tumor cell line’s karyotype, the modal spine cell line karyotype (panel: Sp-1) gained two abnormal number 6 chromosomes (blue arrows), with additional material added to the 6q13 arm (blue arrows), in addition to a loss of abnormal chromosome 3 (blue arrow) and five marker chromosomes (M4, M5, M8, M9, and M11, blue type), as well as a gain of four novel marker chromosomes (M12, M13, and Ms, blue arrows). The two central karyotypes (panels: Lu-2 and Lu-3) in Figure 3 are additional lung cell line karyotypes, with interkaryotype distinctions between these and the modal 1° tumor karyotype indicated by blue arrows and losses of marker chromosomes shown in blue type. In these cases, intrakaryotype differences are indicated by green arrows. Surprisingly, a karyotype of the lung cell line (panel: Lu-2) exhibited an apparent gain of chromosome 8 that was not observed in any of the 1° tumor karyotypes (Figure 1), which indicates that this karyotype was derived from a rare, unobserved 1° tumor subclone that had retained chromosome 8 from the parental cell line (Figure 1). In the lower-right-hand side (panel: Sp-2) is a second spine cell line karyotype with an intrakaryotype distinction indicated with a dark red arrow and interkaryotype differences between spine and 1° tumor karyotypes indicated by blue arrows. It can be noted that all three lung (Lu-1–Lu-3) cell line karyotypes are different from each other, as well as from the two spine (Sp-1 and Sp-2) cell line karyotypes. In addition, the three lung cell karyotypes are distinct from all brain and liver cell line karyotypes, and the two spine cell line karyotypes differ from all the liver cell line karyotypes. However, the non-modal spine karyotype (panel: Sp-2) is identical to the modal brain cell line karyotype (Br-1, Figure 2), which may indicate at least a limited conserved adaptation to different tissue microenvironments. A summary of the differences between each cell line’s modal karyotype relative to the 1° tumor’s modal karyotype is presented in a karyogram (Figure 4), which indicates that although relative to the 1° tumor cell line, substantial amounts of genetic material were altered across several characteristic diploid chromosomes or chromosome regions in the metastatic cell lines, generally, several abnormal chromosomes from the 1° tumor were retained, and most of the divergent genetic changes in the metastatic karyotypes involved large changes in marker chromosome content.

### 3.2. Relative Differential Expression Levels of Proteins/Transcripts Associated with S/G2-Mitosis of the Cell Cycle

To gain a better understanding of factors that could compromise cellular processes of cell division (S/G2-mitosis) and, consequently, be involved in driving/maintaining chromosomal instability (CIN) and, subsequently, aneuploidy, we evaluated changes in the expression levels of the proteins (in our proteomic dataset) of these processes. Table 1 shows the linear fold change (F.C.) in the range ≤−1.25 ≥1.25 of 1° tumor cell line proteins relative to the parental cell line proteins in the cell division processes listed in the Methods section, except for the nuclear envelope regulation process, as no 1° tumor protein levels of this process were found to have changed relative to the parental cell line levels. Chromosomal locations are also given, and notably, despite not being observed in the karyotypes presented in Figure 1, three genes of the proteins (ESCO2, MTBP, and RAD54B) are located on chromosome 8. This is consistent with the finding of a chromosome 8 in one of the lung cell line karyotypes and supports the suggestion that the 1° tumor harbored a subclone that retained this chromosome from the parental cell line and/or that chromosome 8 genetic material was incorporated into one or more of the 1° tumor’s marker chromosomes. Sixty of the 1° tumor proteins were found to be associated with the various biological processes, and of these, 75% exhibited increased levels of expression (Table 1). At the same time, 83% of the 60 proteins exhibited no change in transcript (mRNA) levels. Nonetheless, as indicated in Table 1 (bold type and underlined F.C. values), twelve proteins were associated with their corresponding transcripts. Of these, three (TUBB3, PCLAF, and BLM) had elevated levels of expression, as did their matched proteins, while another three (MTBP, RAD54B, and TYMS) had diminished levels of expression, as did their matched protein counterparts; notably, in all of the six remaining matched transcripts/proteins (PRR11, ZW10, HIST1H3A, HIST1H3C, HIST1H3D, and HIST1H3G), we found that decreased transcript levels were matched to increased protein levels. As such, this mismatch of protein levels and their transcript levels is supporting evidence that a decrease in transcript levels does not necessarily reflect the status of their protein counterpart levels. Overall, the fact that only 17% of proteins could be matched to their transcripts reflects the established differential regulation of the levels of transcripts and the levels of their corresponding proteins, as discussed previously [40]. On balance, relative to the parental cell line, most of the 1° tumor proteins exhibited increased levels of expression in the indicated processes (particularly in responses to DNA damage), which participate in accurate error-free transversion through S/G2-mitosis. Consequently, these results can be interpreted as a measure of decreased CIN or increased stability in the 1° tumor’s accuracy of traversing S/G2 mitosis over that of the parental cell line. 

The analysis of these S/G2 mitosis-associated processes and proteins was extended to a comparison of these proteins in the metastatic cell lines relative to the 1° tumor cell line (Table 2 and Appendix A). As in Table 1, in Table 2, Appendix A, proteins with transcript counterparts are indicated by bold, underlined type and the percentage of these was 22.1% for brain, 23% for liver, 20% for lung, and 25% for spine, which again indicates that the bulk of these proteins were not matched to their transcript counterparts. However, in these cases, all but two (SMC in lung and HMGB1 in brain) were in the same direction of change (increased or decreased levels) as their associated proteins. In these tables, when considering all four metastatic cell lines, beige shading indicates that no transcript counterparts were observed for the proteins in these processes. Appendix A shows relatively moderate-to-low decreases in protein levels in all the biological processes listed in Methods. However, Appendix A shows augmented protein levels, which were found only in centrosome regulation, kinetochore formation, microtubule regulation, DNA damage, DNA replication, and chromosome organization, while no increases in protein levels were found in cell cycle regulation, cytokinesis, chromosome partition, chromosome condensation, nuclear envelope regulation, spindle assembly and regulation, spindle checkpoint, or DNA metabolism. Table 2 shows relatively high levels of diminished protein levels in all biological processes, except for cytokinesis, as well as nuclear envelope regulation, where no proteins were observed to have F.C.s. The total number of proteins for the metastatic cell lines was 136 for brain, 74 for liver, 60 for lung, and 28 for spine. Of these total proteins, the number that were decreased in each metastatic cell line was 104 (76.5%) for brain, 60 (81.1%) for liver, 50 (83.3%) for lung, and 16 (57.1%) for spine. Furthermore, when considering the total number of proteins found in each metastatic cell line, the percentages that were at the higher levels of decreased expression, i.e., those solely represented in Table 2, were similar between the brain and lung cell lines, at 36.8% and 40%, respectively, and lower in the liver and spine cell lines, at 17.6% and 10.7%, respectively. In summary, metastatic cell lines exhibited a decline in the majority of proteins associated with S/G2 mitosis processes, which can be interpreted as a measure of increased CIN of these cell lines’ genomes. Alternatively, although there was a likely decrease in the competence/efficiency of these biological processes in the metastatic cell lines, these cell lines may have acquired an increase in compensatory mechanisms to stabilize mitosis/cytokinesis so as to minimize increased aneuploidy, which could promote cellular survival. 

Along the lines of the analysis described above for Table 2, a protein abundance measure, i.e., percentage of the number of proteins from each metastatic cell line in each of the S/G2 mitosis processes relative to the total number of all proteins in the process, was calculated as an estimate of which of the biological processes may be most impacted by the observed F.C.s (≤−1.25) in protein levels. It was reasoned that higher percentages would likely reflect a higher impairment/dysregulation of the normal functioning of a given biological process. Table 3 indicates that DNA metabolism (yellow shading) was the most likely process to be dysregulated by proteins that are involved in modulating this process. This possible effect was seen across three cell lines (brain, liver, and lung) in instances in which percentages indicate that greater than 40–50% of the pathway would be compromised. We then used roughly 25–30% (green shading) as a minimal cutoff value to decide which other processes might be dysregulated within, as well as across, cell lines. Using these criteria, DNA condensation would likely be dysregulated across the brain, liver, and lung cell lines, with cytokinesis, chromosome partition, microtubule regulation, spindle checkpoint, and DNA replication likely dysregulated in the brain cell line. Using the same metric, we also analyzed the differential regulation of transcript levels for a comparison with protein levels in the S/G2 mitosis processes. Table 4 indicates that a much larger number of transcripts with F.C.s ≤ −1.25 were found in the liver, lung, and spine cell lines compared to the number of proteins (Table 3). Consistent with the findings of the percentage of proteins associated with DNA condensation (Table 3), a large number of transcripts linked to DNA condensation were decreased in liver, lung, and spine. A decrease in transcript levels associated with DNA metabolism in the brain cell line is consistent with the findings of decreased protein levels in this process (Table 3). Notably, the liver cell line exhibited the greatest numbers of decreased levels of transcripts, which could potentially impact all listed biological processes, except for DNA metabolism and chromatin organization (Table 4). Along with DNA condensation, relatively moderate numbers of diminished transcripts of the lung cell line could possibly affect centrosome regulation, chromosome partition, microtubule regulation, DNA damage, and DNA replication. Overall, Table 3 and Table 4 again indicate a differential regulation of protein levels relative to their mRNA counterparts across all cell lines, which was very striking in the liver cell line.

### 3.3. Numbers of Exome-Specific SNPs Differ in Each Chromosome across Isogenic Cell Lines

To determine whether there were gene-level instability differences between the 1° tumor and parental cell lines, as well as between the 1° tumor cell line and each metastatic cell line and between metastatic cell lines, we analyzed the linear fold change in SNP frequencies in each chromosome in each cell line. Figure 5 shows plots of the resulting datasets where linear F.C.s in the frequencies of the 1° tumor cell line/parental cell line or metastatic cell line/1° tumor cell line SNP ratios are plotted and ratios between ≤−1.25 and ≥1.25 are bounded by red lines. Figure 5A indicates that the mean values of the SNP 1° tumor/parental ratios were ≥1.25 for 14 chromosomes (#s: 3, 5, 6, 9–16, 18, 21, and X; Table 5), which indicates that the frequencies of the occurrence of gene-specific SNPs for the majority the 1° tumor genome increased relative to the parental cell line’s genome. Based on this metric, at the gene level, the 1° tumor exhibited increased instability relative to the parental cell line. The notable exception was chromosome 4, where the number of SNPs in the 1° tumor cell line decreased (mean F.C = −1.64, Table 5) relative to the parental cell line, which indicates an increased stability against SNP events. However, there was no change in SNP frequencies in eight chromosomes (#s: 1, 2, 7, 8, 17, 19, 20, and 22; Table 5). Similarly, Figure 5B–E show the linear F.C. (metastatic cell line/1° tumor cell line ratio) in SNP frequencies in the genes of each chromosome for each of the metastatic cell lines, brain, liver, lung, and spine, respectively. The majority of the chromosomes in both the brain and liver cell lines exhibited mean increased SNP ratios (≥1.25, decreased stability), i.e., across 13 (#s: 1, 2–5, 8, 9, 12, 15, 17, 18, 21, and 22; Table 5) and 14 (#s: 2–4, 6, 8, 10, 12, 14–17, 19, 22, and X; Table 5) chromosomes, respectively (Figure 5B,C). No F.C.s in SNP frequencies, i.e., ratios of ~1, were observed for any of the remaining chromosomes in these two cell lines. For the lung cell line, only the genes on chromosome 15 showed an increase in instability (mean F.C. = 1.54 in the number of SNPs) relative to the 1° tumor cell line, while five chromosomes (#s: 7, 11, 20, 21, and X; Table 5) had gene-level increases in stability, i.e., mean linear F.C.s ≤ −1.25. For the spine cell line, five chromosomes (#s: 4–6, 8, and 22; Table 5) had increased mean linear F.C.s in SNPs, i.e., gene-level increases in instability relative to the 1° tumor, and three chromosomes (#s: 13, 18, and 21; Table 5) had decreases in instability relative to the 1° tumor cell line. All other chromosomes in the lung and spine cell lines showed no changes in F.C. ratios for SNPs relative the 1° tumor cell line. It must be emphasized that the F.C.s shown in Figure 5 represent two different sets of separate chromosomal ratios, i.e., 1° tumor cell line/parental cell line ratios and each metastatic cell line/1° tumor cell line ratios, which obscures the findings of the compounded increases in SNP frequencies in metastatic cell line chromosomes above those that occurred in the 1° tumor relative to the parental cell line. Thus, Table 5 shows that many of the increased SNP frequencies in chromosomes (#s: 3, 5, 6, 9, 10, 12, 14, 15, 16, 18, and 21) of the 1° tumor further increased in the metastatic cell lines.

A compilation of the linear F.C. of SNP frequencies for each chromosome in the 1° tumor cell line relative to the parental cell line and metastatic cell lines relative to the 1° tumor cell line is shown in Figure 6, which can be regarded as reflecting the F.C.s of the collective exome SNP frequencies for each cell line and, hence, changes in genomic instability at the level of the exomes. Figure 6 indicates that, on average, relative to the parental cell line, the 1° tumor cell line cell line had gene-level instabilities, i.e., increased F.C. SNP frequencies across its exomes (mean F.C. = 1.36). This was also the case for the brain and liver cell lines’ average increases in F.C.s in exome-wide SNP frequencies (mean F.C.s = 1.38 and 1.36 for brain and liver, respectively) relative to 1° tumor cell line, while, on average, relative to the 1° tumor cell line, the lung and spine cell lines’ exome-wide F.C.s in SNP frequencies remained unchanged (mean F.C.s = 0.93 and 1 for lung and spine, respectively); nevertheless, as indicated above, notable increases, along with decreases in gene-level stabilities, were observed for specific chromosomes of the lung and spine cell lines. Consistent with these results, Table 6 indicates that linear F.C.s in SNP frequency comparisons between the individual chromosomes across cell lines were fewer in the brain and liver cell lines vs. the 1° tumor cell line than for the lung and spine cell line vs. the 1° tumor cell line, reflecting the results presented in Figure 6. Similarly, comparisons between the brain and liver cell lines show only two that were significantly different (Table 6), which is consistent with Figure 6. Table 6 also indicates that 15 of the 23 comparisons between the liver and lung cell lines were significantly different, as well as 10 significant differences in comparisons between the liver and spine cell lines, which is again consistent with Figure 6. However, there were fewer than expected significant differences between the brain and lung (only two differences), as well as between the brain and spine (only three differences); however, this was likely due to the relatively high amount of inherent variance between the biological replicates of the brain dataset. On the other hand, given the similarity between the lung and liver cell line plots in Figure 6, the finding that only a few of the comparisons in Table 6 were significantly different was to be expected.

### 3.4. Chromosomal Level Amplifications, Gains, and Losses

In the present analysis, it became apparent that large portions of the parental cell line’s karyotypes were retained in the1° tumor’s karyotypes, as well as across the metastatic cell line karyotypes. This led us to consider an analysis of a small fraction of conserved yet aberrant chromosomal regions that likely contribute to the successful growth of the 1° tumor, as well as dissemination and growth of metastasis, which could also provide insights into druggable targets across all manifestations of a metastatic disease. As such, Figure 7 shows three such large chromosomal alterations that were retained across all cell lines. An interstitial amplification within chromosome arm 7q increased the copy numbers of 33 genes (Figure 7A). Deviations from the normal diploid copies to three copies of *MNX, XRCC2, KMT2C, CHPF2, and EZH2*; to five copies of *EPHB6, PRSS1, MGAM, BRAF, MET, RINT1*, and *EPHB4*; and to four copies for the other 21 genes (Figure 7A). A search for reports (PubMed) of known activities of these genes in breast cancer showed that only three (*CHPF2, KEL, and CCT6P1*) have not yet been associated with this cancer. Figure 7B shows that a gain in the entire chromosome arm 20p increased the copy number of 17 genes to 3 copies for *GNAS, CD40, PTPRT,* and *MAFB* and to 4 copies for the remaining 13 genes. Only *MAFB* has not been reported in breast cancer. Figure 7C indicates the loss of 6sixgenes due to an interstitial loss of chromosome arm 12p. Consistent with these losses, *PTPRO* and *CDKN1B* have been reported to be tumor suppressor genes [48,49,50], which means that these losses may be advantageous for tumor growth and disease progression. However, the four other genes (*ETNK1, ABCC9, RECQL, and ETV6*) can be upregulated in breast cancer [51,52,53,54]. Given the latter discordant findings, we screened the combined set of genes from all three chromosomal sites against our transcriptomic and proteomic datasets to determine whether the genomic amplifications, gains, and losses were reflected in the transcriptomes and proteomes of these cell lines. We screened for F.C.s between ≤−1.25 and ≥1.25 of the 1° tumor and metastatic cell lines relative to the parental cell line, as well as the metastatic cell lines relative to the 1° tumor cell line. These analyses indicated that a large portion of the genes (amplified, gained, or lost) exhibited differential expression (tissue-context-specific) of transcript counterparts that diverged (increased or decreased) from their amplifications, gains, or losses relative to their genes (Table 7, Table 8 and Table 9). Similarly, we found tissue-specific differential divergences in the proteins of amplified or gained genes (Table 10 and Table 11), as well as low levels of concordance between changes found in transcript F.C.s relative to these genes and the changes found in F.C.s of their protein counterparts (compare Table 7 and Table 8 to Table 10 and Table 11). In addition, some of the amplified genes (Figure 7A), such as *MNX1* (homeodomain family, i.e., developmental gene), PRSS1 (germline-associated gene), *CCT6P1* (pseudogene), and *GRM3*, were not represented in our transcriptome dataset and therefore not recorded in Table 7. The lack of representation of these genes in the transcriptome dataset was largely reflected in our proteome dataset, where MNX1, GRM3, and CCT6P1 were also not found but PRSS1 was represented (Table 10). Moreover, several more amplified or gained genes (Figure 7A,B), such as *XRCC2, KEL, MGAM, SMO, GRM8, PIK3CG, RELN, CD36, MAGI2, SEMA3A, RTEL1, PTPRT, MAFB*, and *ASXL1*, were absent from the proteome dataset, regardless of transcript level. The interstitial loss of genes in chromosome arm 12p (Figure 7C) also exhibited tissue-type-dependent differential F.C.s in the expression of transcripts (Table 9). However, despite increases in some of the levels of expression of the transcripts of these genes, relative to the parental cell line, there was an apparent loss of expression of five of these genes at the protein level; ETNK1, ABBC9, PTPRO, ETV6, and CDN1B proteins were not found in our proteomic dataset, while RECQL was observed but with no changes in expression levels across all tissue types. Overall, these F.C. comparisons proved to be consistent with our previous findings that transcript and protein levels are not generally found to be correlated [40] and extend those results to differential changes in chromosome-level gene expression, regardless of a state of gene amplification, gain, or loss, which reflects compounded complexities due to changes influenced by tissue context.

### 3.5. Gene Variants by DNA-Based NGS 

NGS was performed for the parental cell line, 1° tumor, and metastatic cell lines with mean unique sequencing reads ranging from 1241× to 1485×. A total of 143 variants were found among these cell lines. Of these, 125 variants (87%) were shared across all cell lines in this study, while 18 variants (13%) were presented either only in one cell line or shared in two to five cell lines (Table 12). The parental cell line had five variants: *DDX41, GRIN2A, LILRB1, PLCG1*, and *PCLO*, which were not detected in the 1° tumor or metastatic cell lines (Table 12). *GRIN2A* is a subunit of the NMDA glutamate receptor and is recurrently altered by mutation in various cancer types. The *GRIN2A* E1123* variant, as found in the parental cell line, is likely oncogenic with a likely loss of function. The lung metastatic cell line had *NHS* and *PIK3R1* variants, and the liver metastatic cell line had an *EIF4A1* variant (Table 12). Both liver and spine cell lines had *EPHA2* and *ERCC3* variants (Table 12). The 1° tumor had eight variants in *MKI67, PRKN, PCLO, POLE, CDKN1C, IGSF3*, and *MED12* genes, which were also present in the brain and spine metastatic cell lines but not present in the parental cell line (Table 12). Among these eight variants, the two variants in *IGSF3* and *MED12* were present in the lung and the liver metastatic cell lines, and the three variants in *PCLO, POLE*, and *CDKN1C* were present in the liver metastatic cell line (Table 12).

## 4. Discussion

The focus of our previous multiomics-based studies was to characterize the transcriptomic, proteomic, and metabolomic distinctions of the isogenic cell lines that were generated from a human xenograft model system of metastatic disease in mice [40,41,42]. Our reasoning was that tissue-specific microenvironments drive altered phenotypes as metastatic cells adapt to each organ. A goal was to emphasize that the biological divergence of metastatic lesions from a 1° tumor needs to be considered for the development of more efficacious treatments against deadly metastasis. Here, by studying karyotypes; biological processes implicated in CIN; SNPs; losses, gains, and amplifications of chromosomal regions; and gene mutation variants across these cell lines, our focus was to expand our understanding of molecular and biological distinctions that exist between tissue-specific metastatic cell lines and their divergence from the 1° tumor cell line, as well as from each other. 

Cytogenomic studies in clinical settings have consistently demonstrated that copy number variations, ploidy, chromosomal aberrations, and heterogeneity are very often independent prognostic markers of survival and resistance to chemotherapies [4,25,28,30,55,56]. Moreover, an “aneuploidy score” was recently proposed; a high aneuploidy score is associated with a poor outcome in patients undergoing immunotherapy [57]. Nevertheless, most aneuploidy assessments have been performed on primary tumor samples. Although a few studies have reported a comparison of the cytogenomics of primary tumors and a metastatic site [9,58], very few studies have made comparisons across two or more metastatic sites [38]. Within this context, our human xenograft metastatic model system in mice provided us with the ability to assess the aneuploidies of four metastatic cell lines that were generated from specific organs (brain, liver, lung, and spine) and make comparisons of aneuploidies between these cell lines, as well as to aneuploidies of the 1° tumor cell line. Given that implanting parental cells into the mammary fat pad of a mouse drastically changes growth conditions relative to those of cell culture, we began with a cytogenomic comparison between the parental cell line grown in culture and the 1° tumor cell line (Figure 1). This revealed that both cell lines exhibited several different karyotypes and that the 1° tumor cell line had diverged from the parental cell line, with numerical aberrations in the form of gains and losses of entire chromosomes, along with structural aberrations, which, in sum, indicated changes to very large amounts of genetic material. It was also revealed that, although our karyotyping was comprehensive in scope, rare clones were missed, such as a 1° tumor karyotype with a chromosome 8. The latter finding highlights the fact that, due to the vast numbers of cells in a tumor, not every karyotype (clone) can be expected to be directly found and studied, which has implications for the development of therapies that are meant to be broadly effective against all of a tumor’s cells. 

The scope of this complexity increased when found that the processes involved in the progression of metastasis to brain, liver, lung, and spine caused further evolution, which resulted in a variety of organ-specific karyotypes that differed from the 1° tumor (Figure 2 and Figure 3), as well as between each metastatic cell line (Figure 2 and Appendix A). Thus, in concordance with our earlier multiomic datasets, cytogenomic analyses showed that adaptations to different organ microenvironments resulted in substantial intra- and interkaryotype heterogeneity and metastatic karyotypes that diverged from the 1° tumor karyotypes. 

To better understand possible causes of this vast inter- and intrakaryotype heterogeneity, we studied CIN. CIN, the loss of the absolute fidelity of chromosomal replication and segregation during cell division, has been established as the principal cause of aneuploidy [6,10,15]. Several forms of CIN have been characterized, including the chromosome–fusion–bridge cycle [59], centrosome amplification [8,10], kinetochore–microtubule attachment errors [14], replicative instability [11], single “catastrophic events” or punctuated evolution [13,60,61], and chromothripsis [16,17,20,23]. Recognizing that CIN is manifested during cell division (mitosis), we sought to link biological processes associated with S/G2-M phases of mitosis through the proteins that participate in these processes [45]. Thus, we screened an established 469 genes in 14 biological processes [45] against our proteomic dataset and catalogued the proteins that had linear fold changes between ≤−1.25 and ≥1.25 relative to the parental cell line in the case of the 1° tumor or relative to the 1° tumor in the case of the metastatic cell lines. We focused on the proteins rather than the transcripts, as we reasoned that proteins are the functional components of these biological processes and would therefore best reflect their status. As described above, based on aneuploidy, it could be reasoned that the 1° tumor cell line had a robust CIN phenotype, yet at the protein level in the S/G2-M analysis, we found that there was an overall increase in proteins of the S/G2-M biological processes, which is an implicit indication of a decreased CIN (Table 1). Understanding this inconsistency will require future studies, but it can be stated that a change from parental cell culture growth to in vivo 1° tumor growth is a likely a reason (among others) for this disconnect. 

On the other hand, all the metastatic cell lines showed predominant decreases in F.C.s of proteins relative to the 1° tumor cell line across these biological processes (Table 2 and Appendix A). This could be interpreted as an indication of possible increases in CIN in the populations of the metastatic cell lines and relative to five 1° tumor cell line karyotypes. Nevertheless, without more definitive research, one should consider that conclusions from the S/G2-M analyses of increased or decreased CIN may not be accurately reflected in our karyotype datasets or may be biased due to the relatively limited subsets of protein changes identified among the 469 possible protein changes, i.e., a more comprehensive coverage of the proteins associated with these 14 biological processes could result in more balanced results with findings of either no change in S/G2-M stability or decreased or increased CIN. However, it must be noted that it has been shown that CIN can be experimentally generated by perturbing the expression of selected single proteins [59]. Along these lines, Table 2 indicates that two proteins (CNTROB [62] and NCAPG2 [63]) were ~3- and ~2-fold lower, respectively, across all metastatic cell lines relative to the 1° tumor cell line, which may be adequate to increase CIN across all metastatic cell lines. Moreover, five other proteins (TRIP13 [64], ZW10 [65], PRIM1 [66], CDC45 [67], and RFC3) were found to be decreased by ~1.5 to ~2-fold in brain, liver, and lung cell lines (Table 2). Consequently, overall, the cumulative effect of all the decreases in protein levels (Table 2 and Appendix A) would likely substantially increase CIN in the metastatic cell lines relative to the 1° tumor cell line. Importantly, several reports are in concordance with the validity of these S/G2-mitosis/biological process results, i.e., disruptions of several of the biological processes of mitosis does define CIN, which drives aneuploidy, and prognostic, as well as therapeutic, strategies have been proposed based on such findings [10,31,32,33,63,66,67,68,69,70,71].

To gain further insights into the stability of the genomes of the 1° tumor and metastatic cell lines, we analyzed the fold change in SNP frequencies across all chromosomes and the cumulative changes for each cell line’s genome. In the case of the 1° tumor cell line, when considering cumulative changes across all chromosomes, these results indicate a significant average increase in SNP frequencies in the 1° tumor cell line relative to the parental cell line, which demonstrates that controlling factors/processes that modulate SNPs are decreased or compromised in the 1° tumor cell line relative to the parental cell line. Similarly, the brain and liver cell lines exhibited increased instabilities with respect to repairing SNP, causing events such as significant cumulative SNP frequencies exceeding those of the 1° tumor cell line and, by extension, the parental cell line as well. Cumulatively, the frequencies of SNPs in the lung and spine cell line did not change relative to the 1° tumor cell line. In addition, SNP data analyses provided evidence that individual chromosomes have varying degrees of stability toward SNP formation, with the numbers chromosomes and specific chromosomes involved, as well as the amount of change in stability, being a function of tissue type. Thus, we found higher numbers of chromosomes with SNP instability in the 1° tumor, brain, and liver cell lines, while in general, fewer chromosomes with SNP instabilities were found in the lung and spine cell lines. Notably, an increase in SNP stability was infrequently found, i.e., occurring in only one, five, and four chromosomes of the 1° tumor, lung, and spine cell lines, respectively. Our findings of differential tissue-specific distinctions in biological processes implicated in CIN and exome-specific SNP frequencies across cell lines are further indications that tissue-specific biochemical conditions modulate cancer cell evolution during their adaptations to each tissue’s microenvironment. 

Finally, DNA-based NGS revealed different gene variants among the parental cell line, i.e., the 1° tumor cell line, and metastatic cell lines with various gene-variant allele frequencies, which further supports the idea that selection pressures contribute to various organ-specific alterations to the genome populations of the metastases. 

## 5. Conclusions

Karyotyping revealed that our cell lines are not isogenic, i.e., they are instead populations of a variety of related yet distinctly different aberrant karyotypes. The inter- and intrakaryotype heterogeneities that we observed here starkly reflect the well known histological and genetic profiling descriptions of the complex heterogeneity of solid tumors and metastatic lesions (e.g., [72]). These findings indicate that a reason that aneuploidy is associated with poor prognosis and drug resistance is the large distinct subpopulations of cancer cells present in a primary tumor or metastatic lesion. To better understand the ongoing generation of aneuploidy within metastases, we studied changes in the levels of proteins involved in the biological processes of S/G2-M phases of mitosis as a measure of CIN. These results allow us to conclude that these processes are compromised in all the metastatic cell lines relative to the 1° tumor cell line and, in particular, in the brain and liver cell lines. The SNP analyses support this conclusion. 

Overall, our analyses underscore the complexity of tissue-specific differential distinctions between all our cell lines from the level of the genome (i.e., aberrant karyotypes) and gene (differences in SNP signatures) to the transcript and protein levels. This is important to note from the perspective of recent clinical practices aimed at developing targeted treatment regimens. This concept is generally aimed at finding a single or a few druggable targets in a patient’s primary tumor, as it is difficult to find targets that are common to a primary tumor and its metastases due to the divergence of the metastatic cells from their primary tumor, as emphasized here. Notably, our results indicate that even a comprehensive search for such dual lesion targets will miss rare clones, and a proportion of these may be resistant to treatment. Consequently, although it cannot completely solve this problem, our biological process results allow us to suggest some possible pan-metastatic therapeutic targets, i.e., the biological processes that were common to all four or at least three metastatic cell lines: CNTROB (centrosome regulation); NCAPG2 (DNA condensation); TRIP13 and ZW10 (spindle checkpoint); and PRIM1, CDC45, and RFC3 (DNA replication). Moreover, in the case of the brain metastatic cell line, the DNA damage process could be added to this list. Furthermore, the findings of our study of the interstitial amplification within chromosome arm 7q, which was retained across all cell lines (including the 1° tumor cell line), CUX1 and its associated pathways emerged as important therapeutic targets [73,74]. Finally, our biological process results indicate that the DNA damage response processes were generally compromised, which indicates that radiation therapy could represent a complementary component to a chemotherapeutic regime. 

## Figures and Tables

**Figure 1 cancers-15-01420-f001:**
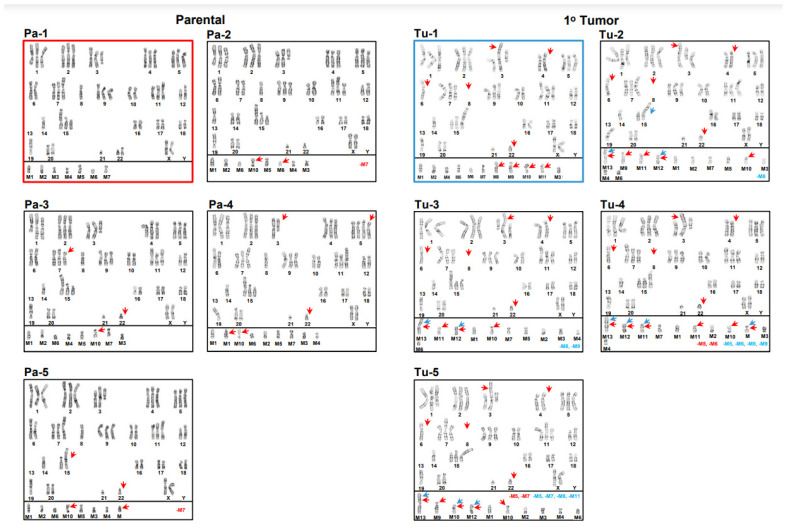
Karyotypes from the parental and 1° tumor cell lines. Five (Pa-1–Pa-5) parental karyotypes are shown on the left-hand side, with the modal karyotype outlined in red, and five 1° tumor karyotypes (Tu-1–Tu-5) are on the right-hand side, with the modal karyotype outlined in blue. Intrakaryotype heterogeneities are indicated by red arrows for the set of four (Pa-2–Pa-5) non-modal parental karyotypes and blue arrows for four (Tu-2–Tu-5) non-modal 1° tumor karyotypes. Relative to the modal parental karyotype, interkaryotype heterogeneities are indicated by red arrows in the five (Tu-1–Tu-5) 1° tumor karyotypes. Relative to the modal parental or modal 1° tumor karyotypes, losses of marker chromosomes in the non-modal karyotypes are indicated by red or blue type, respectively. Modal parental karyotype: 56,XX,+1,der(1;7)(q10;q10),add(1)(q21),+2,+add(3)(q12),+4,+5,dup(6)(p21.3p22),+7,i(7)(q10),-8,+9,inv(9)(q13q?22),+11,add(11)(p14),add(11)(p14),del(12)(p12),-13,-13,-14, +15,add(15)(p11.2),add(18)(p11.2),-19,add(19)(p13),add(20)(q13.2)x2,-21,+mar1,+mar2, +mar3,+mar4,+mar5,+mar6,+mar7. Modal 1° tumor karyotype: 56,XX,+1,der(1;7)(q10;q10),add(1)(q21),+2,add(3)(p25),add(3)(p14),+add(3)(q12),+5,-6, dup(6)(p21.3p22),+7,i(7)(q10),-8,-8,+9,inv(9)(q13q?22),+11,add(11)(p14),add(11)(p14), del(12)(p12),-13,-13,-14,+15,add(15)(p11.2),add(18)(p11.2),-19,add(19)(p13),add(20)(q13.2)x2,-21-22,+mar1,+mar2,+mar3,+mar4,+mar5,+mar6,+mar7, +mar8,+mar9,+mar10,+mar11.

**Figure 2 cancers-15-01420-f002:**
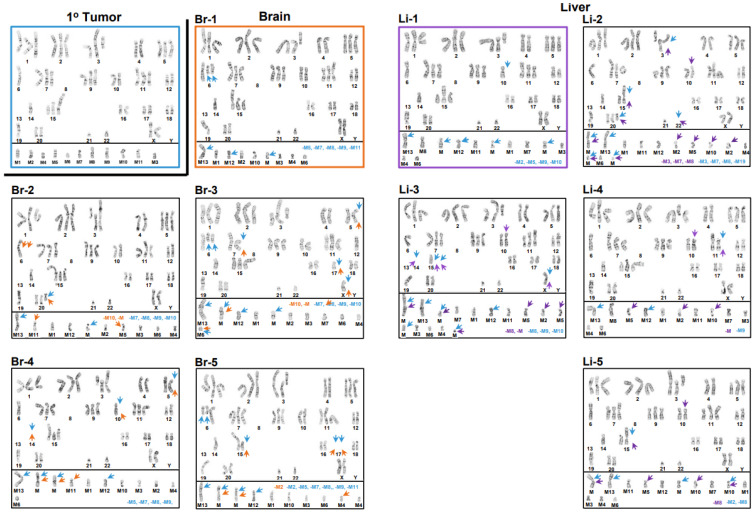
Karyotypes from the brain and liver cell lines compared to the modal 1° tumor karyotype and to each other’s karyotypes. The modal 1° tumor karyotype (upper left-hand corner) is outlined in blue and separated from the five brain (Br-1–Br-5) karyotypes by a black boarder along its bottom and right-hand sides. The modal brain (Br-1) karyotype is outlined in orange. Five liver (Li-1–Li-5) karyotypes are on the right-hand side, with the modal karyotype (Li-1) outlined in violet. Intrakaryotype heterogeneities are indicated by orange arrows for the set of four non-modal brain (Br-2–Br-5) karyotypes and violet arrows for the four non-modal liver (Li-2–Li-5) karyotypes. Interkaryotype heterogeneities between the modal 1° tumor karyotype and both sets of brain and liver karyotypes are indicated by blue arrows. Losses in marker chromosomes between the modal 1° tumor karyotype and the brain and liver karyotypes are indicated with blue type. Similarly, losses in marker chromosomes between the modal brain or modal liver karyotypes and their non-modal karyotype counterparts are indicated by orange or violet type, respectively. Modal 1° tumor karyotype: 56,XX,+1,der(1;7)(q10;q10),add(1)(q21),+2,add(3)(p25),add(3)(p14),+add(3)(q12),+5,-6,dup(6)(p21.3p22),+7,i(7)(q10),-8,-8,+9,inv(9)(q13q?22),+11,add(11)(p14),add(11)(p14),del(12)(p12),-13,-13,-14,+15,add(15)(p11.2),add(18)(p11.2),-19,add(19)(p13),add(20)(q13.2)x2,-21,-22,+mar1,+mar2,+mar3,+mar4,+mar5,+mar6,+mar7,+mar8, +mar9,+mar10,+mar11. Modal brain karyotype: 56,XX,+1,der(1;7)(q10;q10),add(1)(q21),+2,add(3)(p25),add(3)(p14),+add(3)(q12),+5,-6, add(6)(q13)x2,+dup(6)(p21.3p22),+7,i(7)(q10),-8,-8,+9,inv(9)(q13q?22),+11,add(11)(p14), add(11)(p14),del(12)(p12),-13,-13,-14,+15,add(15)(p11.2),add(18)(p11.2),-19,add(19)(p13), add(20)(q13.2)x2,-21,-22,+mar1,+mar2,+mar3,+mar4,+mar6,+mar10,+mar12,+mar13,+mar. Modal liver karyotype: 56,XX,+1,der(1;7)(q10;q10),add(1)(q21),+2,add(3)(p25),add(3)(p14), +add(3)(q12),+5,-6,dup(6)(p21.3p22),+7,i(7)(q10),-8,-8,+9,inv(9)(q13q?22),-10,+11,add(11) (p14),add(11)(p14),del(12)(p12),-13,-13,-14,+15,add(15)(p11.2),add(18)(p11.2),-19,add(19)(p13),add(20)(q13.2)x2,-21,-22,+mar1,+mar3,+mar4,+mar6,+mar7,+mar8,+mar11, +mar12,+mar13,+3mar.

**Figure 3 cancers-15-01420-f003:**
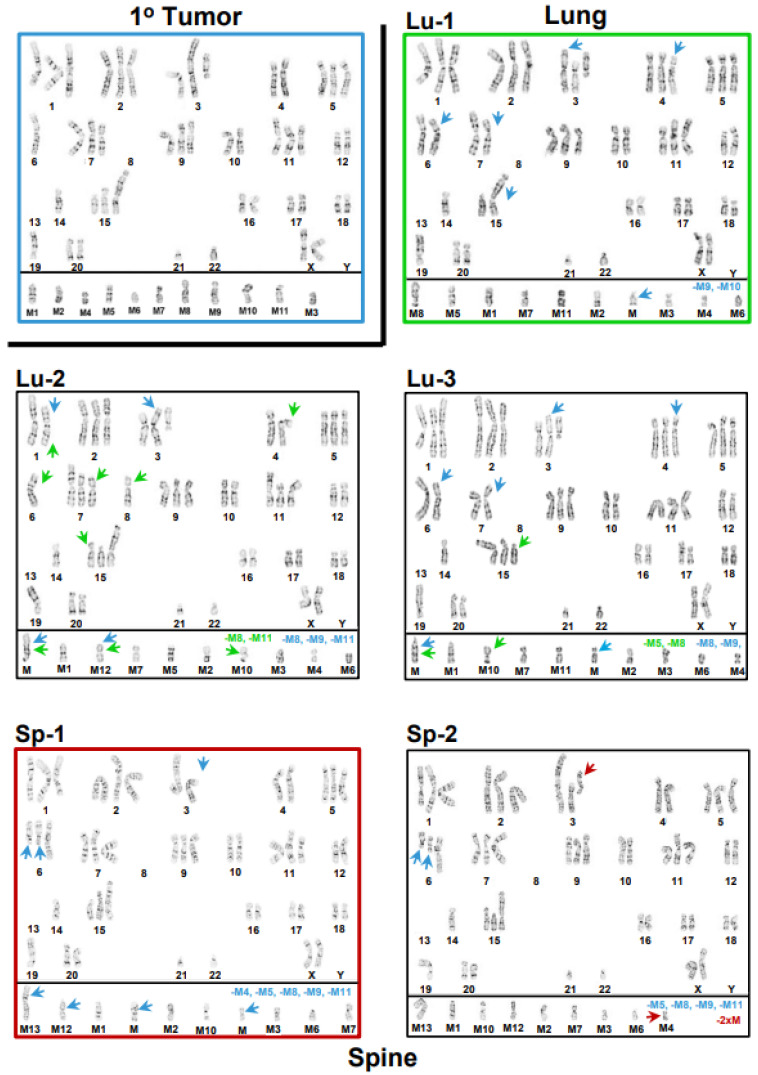
Karyotypes from the lung and spine cell lines compared to the modal 1° tumor karyotype and to each other’s karyotypes. The modal 1° tumor karyotype (upper left-hand corner) is outlined in blue and separated from the three lung karyotypes by a black boarder along its bottom and right-hand sides. The modal lung (Lu-1) karyotype is outlined in green. The two (Lu-2 and Lu-3) centered karyotypes are non-modal lung karyotypes. The two (Sp-1–Sp-2) bottom karyotypes are spine karyotypes, with the modal spine karyotype outlined in dark red. Intrakaryotype heterogeneities are indicated by green arrows for the two non-modal lung (Lu-2 and Lu-3) karyotypes and dark red arrows for the non-modal spine (Sp-2) karyotype. Interkaryotype heterogeneities between the moda1° tumor karyotype and both sets of lung and spine karyotypes are indicated with blue arrows. Losses of marker chromosomes between the modal 1° tumor karyotype and the lung and spine karyotypes are indicated with blue type. Similarly, losses in marker chromosomes between the modal lung or modal spine karyotypes and their non-modal karyotype counterparts are indicated by green or dark red type, respectively. Modal 1° tumor karyotype: 56,XX,+1,der(1;7)(q10;q10),add(1)(q21),+2,add(3)(p25),add(3)(p14),+add(3)(q12),+5,-6,dup(6)(p21.3p22),i(7)(q10),-8,-8,+9,inv(9)(q13q?22),+11,add(11)(p14),add(11)(p14),del(12)(p12),-13,-13,-14,+15,add(15)(p11.2),add(18)(p11.2),-19,add(19)(p13),add(20)(q13.2)x2,-21,-22,+mar1,+mar2,+mar3,+mar4,+mar5,+mar6,+mar7,+mar8,+mar9,+mar10,+mar11. Modal lung karyotype: 55,XX,+1,der(1;7)(q10;q10),add(1)(q21),+2,add(3)(p25),add(3)(p14),+add(3)(q12),+4,+5,dup(6)(p21.3p22),i(7)(q10),-8,-8,+9,inv(9)(q13q?22),+11,add(11)(p14),add(11)(p14),del(12)(p12),-13,-13,-14,add(15)(p11.2),add(18)(p11.2),-19,add(19)(p13),add(20(q13.2)x2,-21,-22,+mar1,+mar2,+mar3,+mar4,+mar5,+mar6,+mar7,+mar8,+mar11,+mar. Modal spine karyotype: 56,XX,+1,der(1;7)(q10;q10),add(1)(q21),+2,add(3)(p25),add(3)(p14),+5,add(6)(q13)x2,+dup(6)(p21.3p22),+7,i(7)(q10),-8,-8,+9,inv(9)(q13q?22),+11,add(11)(p14),add(11)(p14),del(12)(p12),-13,-13,-14,+15,add(15)(p11.2),add(18)(p11.2),-19,add(19)(p13),add(20)(q13.2)x2,-21,-22,+mar1,+mar2,+mar3,+mar6,+mar7,+mar10,+mar12,+mar13, +2mar.

**Figure 4 cancers-15-01420-f004:**
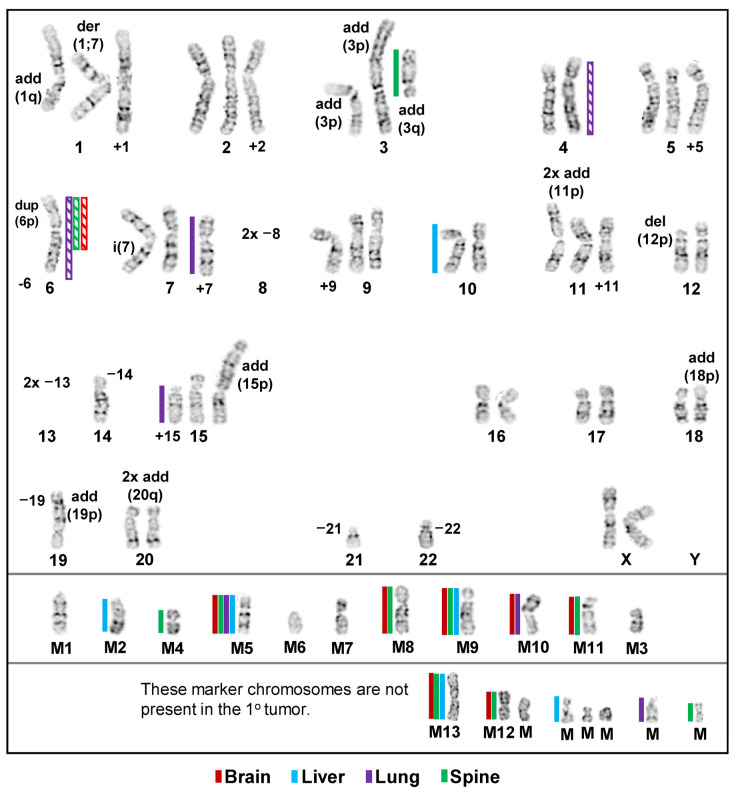
Karyogram of the modal karyotypes of the brain (red), liver (blue), lung (violet), and spine (green) metastatic cell lines compared to the modal 1° tumor karyotype, where the abnormal chromosomes of the latter have been labeled as, e.g., add (1q), der (1;7), etc. Solid colored bars (left of chromosomes) designate losses, and striped bars (right of chromosomes) designate gains.

**Figure 5 cancers-15-01420-f005:**
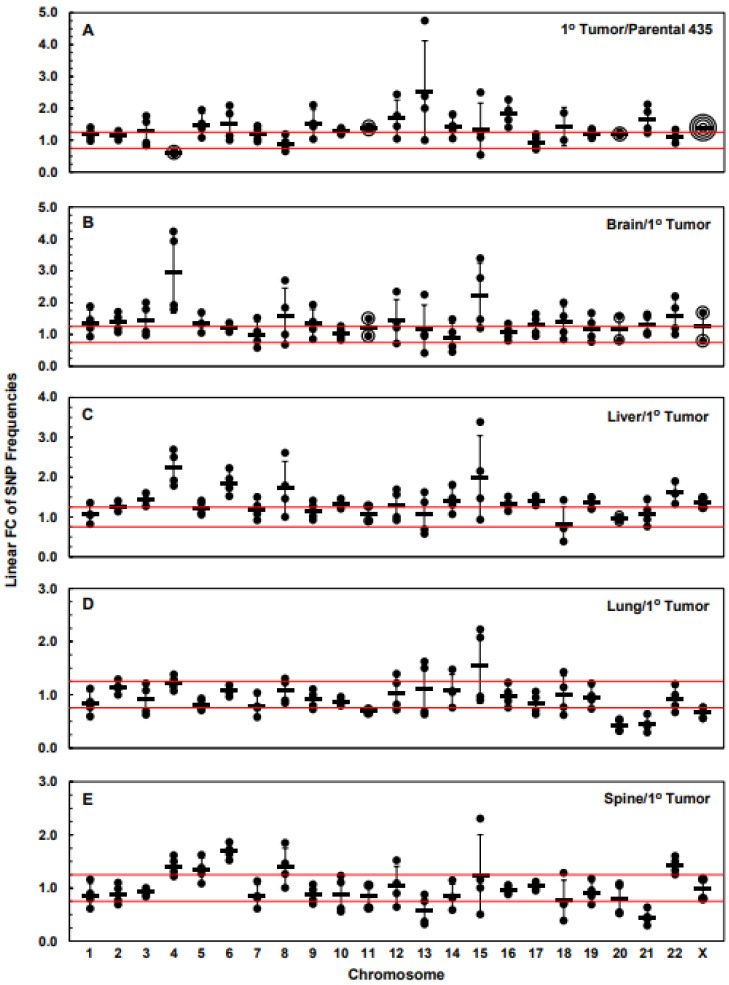
Linear fold changes in SNP frequencies for each chromosome for the 1° tumor cell line relative to the parental cell line (**A**) or for each metastatic cell line relative to the 1° tumor cell line (**B**–**E**). Cutoff boundaries between ≤−1.25 and ≥1.25 F.C. are bounded by red lines. F.C. denotes linear fold change. Rectangular data points = means; error bars = standard deviations.

**Figure 6 cancers-15-01420-f006:**
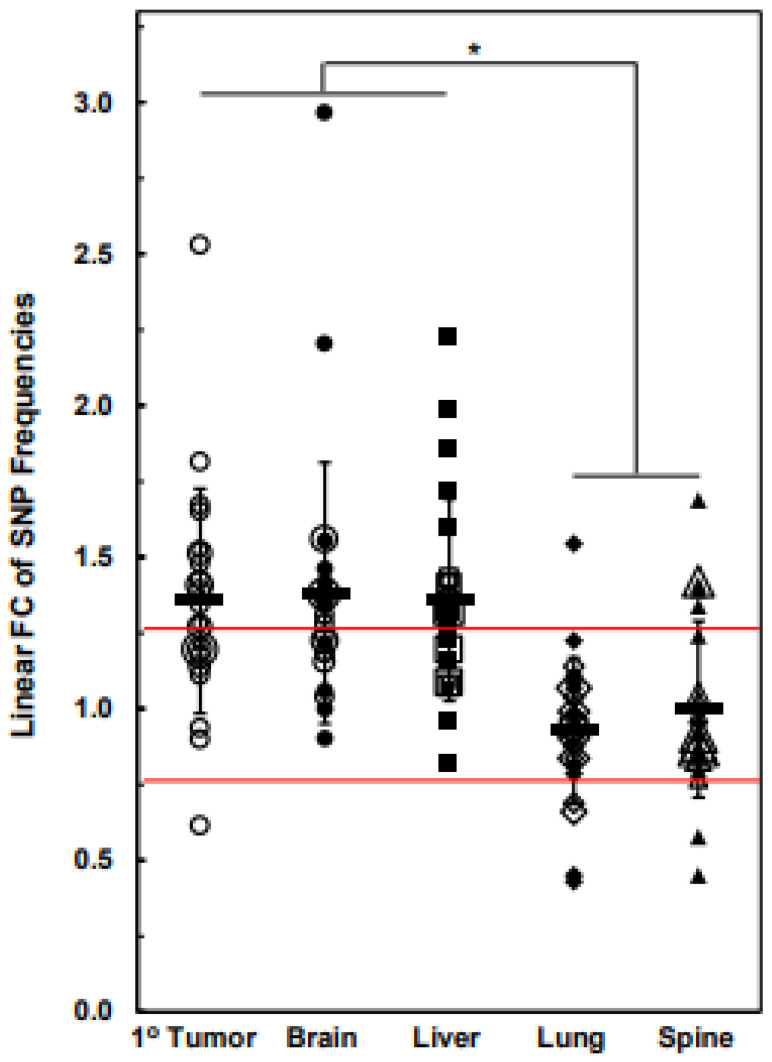
Cumulative linear F.C. of SNP frequencies for all chromosomes of the 1° tumor cell line relative to the parental cell line or for each metastatic cell line relative to the 1° tumor cell line. Cutoff boundaries of between ≤−1.25 and ≥1.25 F.C. are bounded by red lines. F.C. denotes linear fold change. Rectangular data points = means; error bars = standard deviations. * Denotes *p* ≤ 0.001 (two-sided *t*-test).

**Figure 7 cancers-15-01420-f007:**
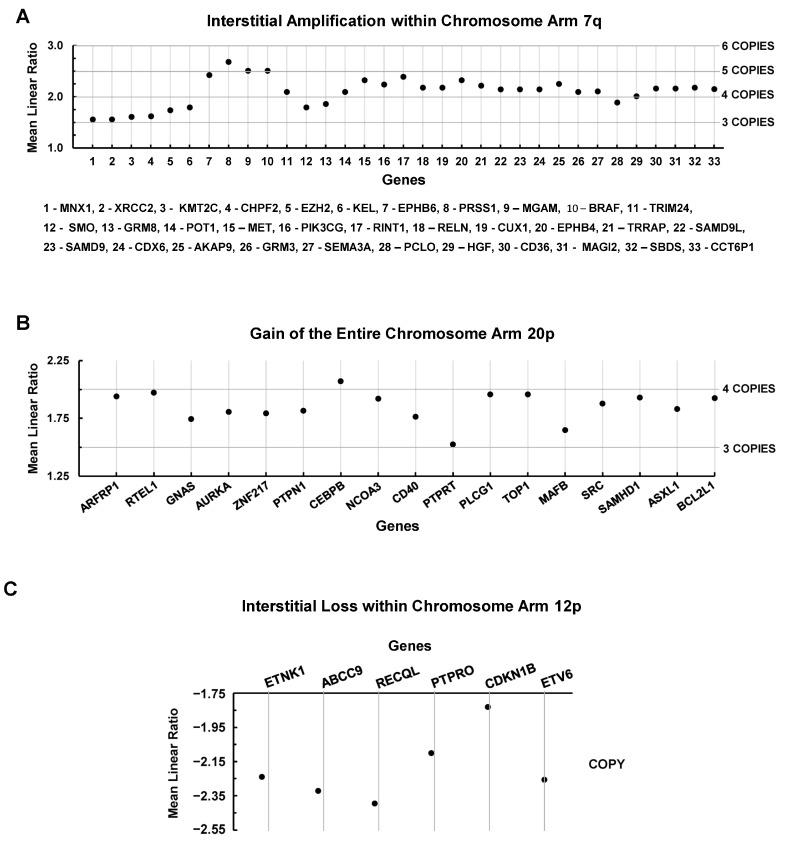
Structural chromosomal abnormalities found across all cell lines. (**A**) Amplification of 33 genes due to an interstitial amplification within chromosome arm 7q. The involved genes were assigned numbers on the *x*-axis, with the key to the gene names listed below the plot. (**B**) Copy number gains of 17 genes due to the gain of the entire chromosome arm 20p. The names of the genes are given on the *x*-axis. (**C**) The loss of six genes due to an interstitial loss within chromosome arm 12p. The names of the involved genes are listed on the *x*-axis. In all three plots, gene copy number changes above or below the normal diploid are shown on the right-hand *y*-axis, which correspond to the mean linear ratios that are scaled on the left-hand *y*-axis.

**Table 1 cancers-15-01420-t001:** Fold change (F.C. ≤−1.25 or ≥1.25) of proteins in the 1° tumor cell line relative to the parental cell line that, if dysregulated, can contribute to CIN/aneuploidy and are consequently impacted by CIN/aneuploidy.

Biological Process	Protein	Description	Chr Location	F.C. 1° Tumor
Cell Cycle Regulation	FZR1	Fizzy/cell division cycle 20 related 1	19p13.3	−1.36
	MELK	Maternal embryonic leucine zipper kinase	9q13.2	1.30
	PRR11	Proline rich 11	17q22	**1.25**
	UBE2C	Ubiquitin conjugating enzyme E2C	20q13.12	−1.38
	UBE2S	Ubiquitin conjugating enzyme E2S	19q13.42	−1.25
Centrosome Regulation	CEP192	Centrosomal protein 192kDa	18p11.21	1.32
	CEP72	Centrosomal protein 72kDa	5p15.33	1.32
	CEP76	Centrosomal protein 76kDa	18p11.21	1.26
	CNTROB	Centrobin, centrosomal BRCA2 interacting protein	17p13.1	3.32
Cytokinesis	ASPM	Abnormal spindle microtubule assembly	1q31.3	1.36
Chromosome Partition	NSL1	NSL1, MIS12 kinetochore complex component	1q32.3	1.73
DNA Condensation	ESCO2	Establishment of sister chromatid cohesion N-acetyltransferase 2	8q21.1	1.39
	NCAPD3	Non-SMC condensin II complex subunit D3	11q25	1.25
	NCAPG2	Non-SMC condensin II complex subunit G2	7q36.3	1.49
Kinetochore Formation	CENPI	Centromere protein I	Xq22.1	1.27
	CENPQ	Centromere protein Q	6p12.3	−1.34
	MIS18A	MIS18 kinetochore protein A	21q22.11	−1.52
	KNTC1	Kinetochore associated 1	12q24.31	1.85
Microtubule Regulation	TUBB3	Tubulin, beta 3 class III	16q24.3	**1.32**
	TUBGCP3	Tubulin, gamma complex associated protein 3	13q24	1.44
	PCNT	Pericentrin	21q22.3	1.67
Spindle Assembly and Regulation	HAUS7	HAUS augmin like complex subunit 7	Xq28	−1.43
Spindle Checkpoint	ZW10	ZW10 kinetochore protein	11q23.2	**1.36**
	ZWILCH	ZWILCH kinetochore protein	15q22.31	1.32
DNA Damage	MTBP	MDM2 binding protein	8q24.12	**−1.32**
	PARP2	Poly(ADP-ribose) polymerase 2	14q11.2	1.24
	TTI1	TELO2 interacting protein 1	20q11.23	1.42
	PCLAF	PCNA clamp associated factor	15q22.31	**1.41**
	DDX11	DEAD/H (Asp-Glu-Ala-Asp/His) box helicase 11	12p11.21	1.32
	POLE	Polymerase (DNA directed), epsilon, catalytic subunit	12q24.33	1.49
	CHEK2	Checkpoint kinase 2	22q12.1	1.37
	PCNA	Proliferating cell nuclear antigen	20p12.3	1.30
	INIP	INTS3 and NABP interacting protein	9q32	−1.35
	NUCKS1	Nuclear casein kinase and cyclin-dependent kinase substrate 1	1q32.1	−1.83
	RAD54B	RAD54 homolog B (S. cerevisiae)	8q22.1	**−1.27**
	BRCA2	Breast cancer 2	13q13.1	1.34
	FANCD2	Fanconi anemia complementation group D2	3p25.3	1.28
	FANCI	Fanconi anemia complementation group I	15q26.1	1.48
DNA Regulation	BLM	Bloom syndrome, RecQ helicase-like	15q26.1	**1.25**
	POLE2	Polymerase (DNA directed), epsilon 2, accessory subunit	14q21.3	1.48
	PRIM1	Primase, DNA, polypeptide 1 (49 kDa)	12q13.3	1.45
	PRIM2	Primase, DNA, polypeptide 2 (58 kDa)	6p11.2	1.26
	CDC45	Cell division cycle 45	22q11.21	1.48
	MCM5	Minichromosome maintenance complex component 5	22q12.3	1.26
	MCM7	Minichromosome maintenance complex component 7	7q22.1	1.31
DNA Metabolism	DTYMK	Deoxythymidylate kinase	2q37.3	1.28
	SLC29A1	Solute carrier family 29 (equilibrative nucleoside transporter), member 1	6p21.1	1.27
	TYMS	Thymidylate synthetase	18p11.32	**−1.40**
Chromatin Organization	HJURP	Holliday junction recognition protein	2q37.1	−1.25
	HIST1H2AC	Histone cluster 1, H2ac	6p22.2	−1.46
	HIST1H3A	Histone cluster 1, H3a	6p22.2	**1.38**
	HIST1H3B	Histone cluster 1, H3b	6p22.2	1.38
	HIST1H3C	Histone cluster 1, H3c	6p22.1	**1.38**
	HIST1H3D	Histone cluster 1, H3d	6p22.1	**1.38**
	HIST1H3F	Histone cluster 1, H3f	6p22.2	1.38
	HIST1H3G	Histone cluster 1, H3g	6p22.2	**1.38**
	HIST1H3I	Histone cluster 1, H3i	6p22.2	1.38
	HIST1H3J	Histone cluster 1, H3j	6p22.2	1.38
	HIST2H2AA4	Histone cluster 2 H2A family member a4	1q21.2	−1.54

**Table 2 cancers-15-01420-t002:** Fold change (F.C. ≤ −1.5) of proteins in metastatic cell lines relative to the 1° tumor cell line that, if dysregulated, contribute to CIN/aneuploidy and are consequently impacted by CIN/aneuploidy.

Biological Process	Protein	Description	Chr Location	F.C. Brain	F.C. Liver	F.C. Lung	F.C. Spine
Cell Cycle Regulation	CDK4	Cyclin-dependent kinase 4	12q14.1	−1.51			
	MELK	Maternal embryonic leucine zipper kinase	9q13.2	**−1.51**			
	PKMYT1	Protein kinase, membrane associated tyrosine/threonine 1	16p13.3	−1.68			
	RBL1	Retinoblastoma-like 1	20q11.23	**−1.62**			
	UBE2C	Ubiquitin conjugating enzyme E2C	20q13.12	−1.59			
Centrosome Regulation	CCT3	Chaperonin containing TCP1, subunit 3 gamma	1q22	−1.65		−1.82	−1.60
	CEP192	Centrosomal protein 192kDa	18p11.21	−1.56		**−1.97**	
	CEP72	Centrosomal protein 72kDa	5p15.33	**−1.67**			
	CNTROB	Centrobin, centrosomal BRCA2 interacting protein	17p13.1	−2.70	−3.45	−3.03	**−3.33**
	KIF24	Kinesin family member 24	9p13.3	−1.96			
Chromosome Partition	MIS18BP1	MIS 18 binding protein 1	14q21.2	−2.05			
DNA Condensation	ESCO2	Establishment of sister chromatid cohesion N-acetyltransferase 2	8q21.1	**−1.50**			
	NCAPG2	Non-SMC condensing II complex subunit G2	7q36.3	**−2.86**	**−1.97**	**−2.46**	**−1.86**
	SMC2	Structural maintenance of chromosomes 2	9q31.1			**−1.57**	
Kinetochore Formation	CENPE	Centromere protein E	4q24	−1.55			
	CENPF	Centromere protein F	1q41	−1.53			
	KNTC1	Kinetochore associated 1	12q24.31			**−1.96**	
Microtubule Regulation	DIAPH3	Diaphanous related formin 3	13q21.2	−1.54			
	KIF4A	Kinesin family member 4A	Xq13.1	−1.52			
	SKA3	Spindle and kinetochore associated complex subunit 3	13q12.11	−2.00			
	PCNT	Pericentrin	21q22.3			−1.52	
Spindle Assembly and Regulation	KPNB1	Karyopherin (importin) beta 1	17q21.32			−1.58	
	HAUS7	HAUS augmin like complex subunit 7	Xq28	−1.52		−1.74	
Spindle Checkpoint	TRIP13	Thyroid hormone receptor interacter 13	5p15.33	**−1.68**	**−1.82**	−1.50	
	ZW10	ZW10 kinetochore protein	11q23.2	−1.87	−1.92	−1.78	
	ZWILCH	ZWILCH kinetochore protein	15q22.31	**−1.67**			
DNA Damage	RAD18	RAD18, E3 ubiquitin protein ligase	3p25.3	−1.56			
	TTI1	TELO2 interacting protein 1	20q11.23		−1.67	−2.02	
	PCLAF	PCNA clamp associated factor	15q22.31	**−3.57**		**−1.54**	
	DDX11	DEAD/H (Asp-Glu-Ala-Asp/His) box helicase 11	12p11.21	−1.54			
	CLSPN	Claspin	1q34.3	**−1.76**			
	TIMELESS	Timeless circadian clock	12q13.3	−1.57			
	TIPIN	TIMELESS interacting protein	15q22.31			−1.66	
	NUCKS 1	Nuclear casein kinase & cyclin-dependent kinase substrate 1	1q32.1			−1.81	
	RAD51AP1	RAD51 associated protein 1	12p13.32	**−1.70**		**−1.48**	
	TSN	Translin	2q14.3		−1.56		
	CHEK1	Checkpoint kinase 1	11q24.2	−1.72			
	ATRIP	ATR interacting protein	3p21.31	−1.60			
	UBE2T	Ubiquitin conjugating enzyme E2T	1q32.1	−1.60			
DNA Replication	BAZ1B	Bromodomain adjacent to zinc finger domain 1B	7q11.23	−1.67			
	HAT1	Histone acetyltransferase 1	2q31.1	**−1.78**	−1.51		
	RMI1	RecQ mediated genome instability 1	9q21.32	**−1.61**			
	POLA2	Polymerase (DNA directed), α2, catalytic subunit	11q13.1	**−1.51**			
	POLE2	Polymerase (DNA directed), ε2, catalytic subunit	14q21.3	**−1.71**			
	PRIM1	Primase, DNA, polypeptide 1 (49 kDa)	12q13.3	−2.02	**−1.90**	−1.52	
	CDC45	Cell division cycle 45	22q11.21	**−2.11**	**−1.54**	**−1.64**	
	MCM10	Minichromosome maintenance 10 replication initiation factor	10p13	**−1.62**			
	CDC6	Cell division cycle 6	17q21.2	**−1.70**			
	FAM111A	Family with sequence similarity 111 member A	11q12.1	−1.64			
	RFC3	Replication factor C sununit 3	13q13.2	−1.69	**−1.77**	−1.68	
DNA Metabolism	DCK	Deoxycytidine kinase	4q13.3		−1.64		
	DHFR	Dihydrofolate reductase	5q14.1	**−2.20**			
	DUT	Deoxyuridine triphosphatase	15q21.1			−1.50	
	RRM1	Ribnucleotide reductase M1	11p15.4	**−1.50**			
	RRM2	Ribonucleotide reductase M2	2q25.1	−2.63		−1.68	
	SLC29A1	Solute carrier family 29 (equilibrative nucleoside transporter), member 1	6p21.1	−1.78			
	TK1	Thymidine kinase 1, soluble	17q25.3	−1.54		−1.65	
	TYMS	Thymidylate synthetase	18p11.32	**−2.39**		−1.58	
Chromatin Organization	ANP32E	Acidic nuclear phosphoprotein 32 family member E	1q21.2		−1.52		
	UHRF1	Ubiquitin-like with PHD & ring finger domains 1	19p13.3	−2.13			
	HIST2H3A	Histone cluster 2, H3a	1q21.2			−1.62	

Yellow–beige shadings indicate that no gene counterparts were observed for the proteins in these processes.

**Table 3 cancers-15-01420-t003:** Protein (F.C.s ≤ −1.25) abundance in each biological process across the four metastatic cell lines.

	Proteins	Brain		Liver		Lung		Spine	
Biological Process	In Each Process	# Proteins	%	# Proteins	%	# Proteins	%	# Proteins	%
Cell Cycle Regulation	68	15	22.1	4	5.9	5	7.4	1	1.5
Centrosome Regulation	31	6	19.4	3	9.7	3	9.7	4	12.9
Cytokinesis	20	6	30.0	0	0.0	2	10.0	0	0.0
Chromosome Partition	25	7	28.0	1	4.0	0	0.0	0	0.0
DNA Condensation	11	5	45.5	3	27.3	3	27.3	2	18.2
Kinetochore Formation	25	6	24.0	1	4.0	2	8.0	1	4.0
Microtubule Regulation	23	7	30.4	0	0.0	3	13.0	0	0.0
Nuclear Envelope Regulation	18	2	11.1	2	11.1	3	16.7	1	5.6
Spindle Assembly and Regulation	20	3	15.0	2	10.0	2	10.0	1	5.0
Spindle Checkpoint	14	7	50.0	3	21.4	3	21.4	2	14.3
DNA Damage	92	21	22.8	10	10.9	11	12.0	2	2.2
DNA Replication	50	14	28.0	8	16.0	4	8.0	1	2.0
DNA Metabolism	17	10	58.8	8	47.1	7	41.2	1	5.9
Chromatin Organization	55	1	1.8	14	25.5	1	1.8	0	0.0

# Denotes number, e.g., 15 Cell Cycle Regulation proteins out of a possible total of 68 were found associated with brain metastases. Percentages ≥40% are shaded yellow, and those ≥25% are shaded green.

**Table 4 cancers-15-01420-t004:** Gene/transcript (F.C.s ≤ −1.25) abundance in each biological process across the four metastatic cell lines.

	Proteins	Brain		Liver		Lung		Spine	
Biological Process	In Each Process	# Proteins	%	# Proteins	%	# Proteins	%	# Proteins	%
Cell Cycle Regulation	68	9	13.2	25	36.8	19	27.9	14	20.6
Centrosome Regulation	31	3	9.7	21	67.7	11	35.5	7	22.6
Cytokinesis	20	2	10.0	9	45.0	5	25.0	0	0.0
Chromosome Partition	25	3	12.0	15	60.0	8	32.0	5	20.0
DNA Condensation	11	2	18.2	10	90.9	5	45.5	5	45.5
Kinetochore Formation	25	5	20.0	14	56.0	6	24.0	8	32.0
Microtubule Regulation	23	1	4.3	12	52.2	7	30.4	7	30.4
Nuclear Envelope Regulation	18	2	11.1	8	44.4	4	22.2	2	11.1
Spindle Assembly and Regulation	20	1	5.0	9	45.0	5	25.0	4	20.0
Spindle Checkpoint	14	5	35.7	6	42.9	3	21.4	2	14.3
DNA Damage	92	16	17.4	48	52.2	30	32.6	17	18.5
DNA Replication	50	16	32.0	36	72.0	19	38.0	13	26.0
DNA Metabolism	17	5	29.4	0	0.0	1	5.9	4	23.5
Chromatin Organization	55	1	1.8	6	10.9	6	10.9	3	5.5

# Denotes number, e.g., 15 Cell Cycle Regulation proteins out of a possible total of 68 were found associated with brain metastases. Percentages ≥40% are shaded yellow, and those ≥25% are shaded green.

**Table 5 cancers-15-01420-t005:** Mean linear F.C. of SNP frequencies for each metastatic cell line relative to the 1° tumor cell line for each chromosome.

Chr	1° Tumor	Brain	Liver	Lung	Spine
**1**	1.18	1.37	1.07	−1.20	−1.16
**2**	1.14	1.38	1.27	1.14	−1.13
**3**	1.27	1.46	1.43	−1.11	−1.09
**4**	−1.64	2.97	2.23	1.22	1.41
**5**	1.48	1.35	1.23	−1.23	1.34
**6**	1.51	1.22	1.86	1.07	1.69
**7**	1.20	1.00	1.20	−1.27	−1.18
**8**	0.90	1.55	1.72	1.07	1.39
**9**	1.52	1.34	1.16	−1.11	−1.14
**10**	1.28	1.04	1.33	−1.14	−1.14
**11**	1.38	1.23	1.09	−1.45	−1.19
**12**	1.67	1.42	1.30	1.04	1.04
**13**	2.53	1.15	1.07	1.11	−1.73
**14**	1.41	0.90	1.42	1.09	−1.19
**15**	1.34	2.21	1.98	1.54	1.24
**16**	1.82	1.06	1.32	−1.02	−1.03
**17**	0.94	1.28	1.41	−1.20	1.03
**18**	1.43	1.37	0.82	−1.01	−1.30
**19**	1.21	1.19	1.35	−1.04	−1.09
**20**	1.20	1.18	0.96	−2.33	−1.25
**21**	1.65	1.31	1.09	−2.23	−2.22
**22**	1.11	1.56	1.60	−1.09	1.42
**X**	1.39	1.24	1.36	−1.52	−1.02

Values with linear F.C.s ≤−1.25 and ≥1.25 are highlighted in gold and green, respectively.

**Table 6 cancers-15-01420-t006:** Significant differences (*p* < 0.05) in linear F.C.s in SNP frequencies in the indicated cell line comparisons across all chromosomes.

Chr	Br vs. 1° T	Li vs. 1° T	Lu vs. 1° T	Sp vs. 1° T	Br vs. Li	Br vs. Lu	Br vs. Sp	Li vs. Lu	Li vs. Sp	Lu vs. Sp
**1**	-----	-----	-----	-----	-----	** 0.06 **	-----	-----	-----	-----
**2**	-----	-----	-----	-----	-----	-----	0.03	-----	0.005	** 0.06 **
**3**	-----	-----	-----	-----	-----	-----	-----	0.015	0.0005	-----
**4**	0.04	0.003	0.001	0.003	-----	-----	-----	0.01	0.01	-----
**5**	-----	-----	0.01	-----	-----	0.009	-----	0.008	-----	0.006
**6**	-----	-----	-----	-----	0.008	-----	0.002	0.008	-----	0.0004
**7**	-----	-----	0.04	-----	-----	-----	-----	0.04	-----	-----
**8**	-----	-----	-----	-----	-----	-----	-----	-----	-----	-----
**9**	-----	-----	0.04	0.03	-----	-----	-----	-----	-----	-----
**10**	-----	-----	0.0004	-----	0.04	-----	-----	0.0004	0.04	-----
**11**	-----	-----	0.004	0.005	-----	0.04	-----	0.04	-----	-----
**12**	-----	-----	-----	-----	-----	-----	-----	-----	-----	-----
**13**	-----	-----	-----	-----	-----	-----	-----	-----	-----	-----
**14**	-----	-----	-----	0.03	-----	-----	-----	-----	0.02	-----
**15**	-----	-----	-----	-----	-----	-----	-----	-----	-----	-----
**16**	0.02	0.05	0.008	0.02	-----	-----	-----	0.035	0.007	-----
**17**	-----	0.009	-----	-----	-----	** 0.06 **	-----	0.002	0.001	-----
**18**	-----	-----	-----	-----	-----	-----	-----	-----	-----	-----
**19**	-----	-----	-----	** 0.06 **	-----	-----	-----	0.015	0.02	-----
**20**	-----	-----	0.00002	-----	-----	0.04	-----	0.0003	-----	-----
**21**	-----	0.002	0.006	0.006	-----	0.01	0.003	0.04	0.009	-----
**22**	-----	0.01	-----	0.04	-----	-----	-----	0.006	-----	0.01
**X**	-----	-----	0.001	0.03	-----	-----	-----	0.0003	0.02	0.04

Abbreviations: Chr–chromosome; 1° T–1° tumor; Br–brain; Li–liver; Lu–lung; Sp–spine. Bold red type indicates a trend of significance.

**Table 7 cancers-15-01420-t007:** Transcripts of the genes from the interstitial amplified region of 7q with differential F.C.s between ≤−1.25 and ≥1.25 relative to the parental or 1° tumor cell lines.

Transcript	1° T/ PCL	Brain/ PCL	Liver/ PCL	Lung/ PCL	Spine/ PCL	Brain/ 1° T	Liver/ 1° T	Lung/ 1° T	Spine/ 1° T
KMT2C	−1.41	1.56	1.36	1.30	1.33	----- ^†^	-----	-----	-----
CHPF2	-----	-----	1.29	-----.	-----	-----	1.35	-----	-----
EZH2	-----	-----	−1.28	-----	-----	-----	−1.30	-----	-----
KEL	-----	2.12	1.30	-----	-----	2.11	1.37	-----	1.26
EPHB6	−1.39	2.02	-----	-----	-----	-----	−1.34	-----	-----
MGAM	-----	−1.41	-----	−1.56	2.08	-----	1.35	−1.43	-----
BRAF	-----	-----	-----	1.51	-----	-----	-----	1.40	-----
SMO	3.76	−6.25	-----	−3.85	−3.70	−1.43	2.15	-----	-----
GRM8	-----.	-----.	-----.	-----.	-----.	N.I.L. ^‡^	N.I.L.	-----.	-----.
POT1	-----	-----	-----	-----	1.27	-----	-----	-----	-----
MET	-----	-----	-----	1.49	-----	-----	-----	1.46	-----
PIK3CG	-----	N.I.L.	-----	N.I.L.	N.I.L.	-----	-----	-----	-----
RINT1	-----	-----	-----	-----	-----	-----	−1.37	-----	-----
RELN	-----	1.78	-----	-----	1.27	1.33	-----	-----	-----
CUX1	−1.30	1.33	1.26	1.30	1.40	-----	-----	-----	-----
EPHB4	-----	1.43	1.46	-----	-----	-----	-----	−1.46	-----
TRRAP	-----	-----	-----	1.36	-----	-----	-----	1.26	-----
SAMD9L	−2.00	1.26	2.60	1.32	1.90	−1.54	1.28	-----	1.26
SAMD9	−1.35	-----	1.52	1.30	1.70	−1.30	-----	-----	-----
CDK6	−1.33	1.37	1.26	1.68	1.37	-----	-----	1.27	-----
AKAP9	-----	-----	-----	1.38	-----	-----	-----	-----	-----
PCLO	−4.17	6.35	2.13	1.76	3.78	1.51	−1.26	−2.27	-----
HGF	N.I.L.	N.I.L.	-----	N.I.L.	N.I.L.	N.I.L.	-----	N.I.L.	N.I.L.
CD36	1.38	-----	-----	-----	-----	-----	-----	1.32	-----
MAGI2	-----	-----	-----	-----	1.31	-----	-----	-----	1.27
SBDS	-----	−1.39	-----	1.34	-----	-----	-----	1.62	-----
SEMA3A	-----	-----	-----	1.89	-----	−1.26	-----	1.65	−1.27

^†^ Denotes no change in expression relative to either the parental (PCL) or 1° tumor (1° T) cell line. ^‡^ Denotes not in the list, meaning not found in our RNA-seq transcript dataset (list). XRCC2 and TRRAP were not included because there was no F.C. in their levels of expression in any of the comparisons.

**Table 8 cancers-15-01420-t008:** Transcripts of the genes from the gain of entire chromosome 20p arm with differential F.C.s between ≤−1.25 and ≥1.25 relative to the parental or 1° tumor cell lines.

Transcript	1° T/ PCL	Brain/ PCL	Liver/ PCL	Lung/ PCL	Spine/ PCL	Brain/ 1° T	Liver/ 1° T	Lung/ 1° T	Spine/ 1° T
ARFRP1	----- ^†^	-----	-----	-----	-----	-----	-----	1.29	-----
RTEL1	-----	-----	-----	-----.	−1.39	-----	-----	-----	−1.38
GNAS	1.33	−1.35	-----	-----	-----	-----	-----	-----	-----
AURKA	-----	−1.39	−1.43	−1.30	-----	-----	-----	-----	-----
ZNF217	-----	-----	-----	1.27	-----	-----	-----	-----	-----
PTPN1	-----	-----	-----	-----	-----	-----	-----	1.25	-----
CEBPB	-----	-----	-----	1.78	-----	-----	-----	2.06	-----
NCOA3	-----	-----	-----	1.59	-----	-----	-----	1.74	-----
CD40	-----	-----	1.27	-----	1.25	-----	-----	−1.42	-----
PTPRT	-----	1.40	-----	N.I.L. ^‡^	-----	-----	-----	N.I.L.	-----
PLCG1	-----	-----	1.37	-----	-----	-----	-----	-----	-----
TOP1	-----	-----	-----	1.54	-----	-----	-----	1.56	-----
MAFB	-----	1.61	1.25	-----	1.70	1.34	−1.48	−1.40	1.42
SRC	-----	-----	−1.30	-----	−1.35	-----	N.I.L.	−1.46	-----
SAMHD1	-----	-----	1.40	-----	-----	1.27	-----	-----	-----
BCL2L1	-----	-----	1.75	1.27	1.70	-----	1.68	-----	-----

^†^ Denotes no change in expression relative to either the parental (PCL) or 1° tumor (1° T) cell line. ^‡^ Denotes not in the list, meaning not found in our RNA-seq transcript dataset (list). ASXL1 is not included because there was no F.C. in its levels of expression in any of the comparisons.

**Table 9 cancers-15-01420-t009:** Transcripts of the genes from the interstitial loss of 12p with differential F.C.s between ≤−1.25 and ≥1.25 relative to the parental or 1° tumor cell lines.

Transcript	1° T/ PCL	Brain/ PCL	Liver/ PCL	Lung/ PCL	Spine/ 435	Brain/ 1° T	Liver/ 1° T	Lung/ 1° T	Spine/ 1° T
ABCC9	----- ^†^	1.25	-----	1.26	-----	-----	-----	-----	-----
RECQL	-----	-----	1.37	1.31	-----	−1.25	−1.33	1.29	-----
PTPRO	-----	-----	-----	1.39	-----	-----	-----	1.53	-----
CDKN1B	-----	-----	1.26	-----	-----	-----	1.31	-----	-----

^†^ Denotes no change in expression relative to either the parental (PCL) or 1° tumor (1° T) cell line. ETNK1 and ETV6 were not included because there was no F.C. in their levels of expression in any of the comparisons.

**Table 10 cancers-15-01420-t010:** Proteins of the genes from the interstitial amplified region of 7q with differential F.C.s between ≤−1.25 and ≥1.25 relative to the parental or 1° tumor cell line.

Transcript	1° T/ PCL	Brain/ PCL	Liver/ PCL	Lung/ PCL	Spine/ PCL	Brain/ 1° T	Liver/ 1° T	Lung/ 1° T	Spine/ 1° T
KMT2C	----- ^†^	−1.48	-----	−1.32	−1.27	−1.29	-----	-----	-----
EZH2	-----	-----	-----	-----	-----	−1.33	-----	-----	-----
EPHB6	-----	-----	-----	-----	-----	-----	-----	1.42	1.32
PRSS1	1.30	-----	-----	-----	-----	−1.25	-----	-----	-----
BRAF	-----	-----	-----	1.51	-----	-----	-----	1.40	-----
TRIM24	-----	-----	-----	-----	1.33	-----	-----	-----	-----
MET	-----	−1.31	-----	-----	-----	−1.25	-----	-----	-----
RINT1	1.40	-----	-----	-----	-----	−1.47	−1.45	−1.32	-----
CUX1	-----	1.35	-----	-----	-----	1.40	-----	-----	-----
SAMD9L	2.14	1.37	1.64	2.13	1.42	−1.56	−1.31	-----	−1.52
SAMD9	1.63	-----	1.76	2.02	1.72	−1.56	-----	-----	-----
CDK6	1.30	-----	-----	-----	-----	−1.43	−1.37	-----	-----
PCLO	5.20	4.45	6.87	2.90	4.25	-----	1.32	−1.79	-----
HGF	2.73	1.90	3.48	1.43	1.42	−1.47	1.25	−1.96	−1.33
SBDS	-----	−1.37	−1.61	-----	-----	−1.39	−1.64	-----	-----

^†^ Denotes no change in expression relative to either the parental (PCL) or 1° tumor (1° T) cell line. CHPF2, POT1, EPHB4, TRRAP, and AKAP9 were not included because there was no F.C. in their levels of expression in any of the comparisons.

**Table 11 cancers-15-01420-t011:** Proteins of the genes from the gain of the entire chromosome 20q arm with differential F.C.s between ≤−1.25 and ≥1.25 relative to the parental or 1° tumor cell line.

Transcript	1° T/PCL	Brain/PCL	Liver/ PCL	Lung/ PCL	Spine/ PCL	Brain/ 1° T	Liver/ 1° T	Lung/ 1° T	Spine/ 1° T
GNAS	----- ^†^	−1.26	−1.37	−1.43	−1.37	-----	-----	-----	-----
AURKA	-----	−1.37	-----	-----.	-----	−1.29	-----	-----	-----
ZNF217	-----	-----	−1.28	-----	-----	-----	−1.32	-----	-----
NCOA3	-----	-----	-----	-----	-----	-----	-----	1.25	-----
PLCG1	-----	1.38	-----	-----	-----	-----	-----	-----	-----
SRC	-----	-----	-----	-----	-----	-----	-----	1.29	-----
SAMHD1	-----	−1.28	-----	1.60	-----	−1.56	-----	1.30	-----
BCL2L1	-----	1.27	-----	−1.25	-----	-----	-----	-----	−1.38

^†^ Denotes no change in expression relative to either the parental (PCL) or 1° tumor (1° T) cell line. PTPN1, CEBPB, CD40, and TOP1 were not included because there was no F.C. in their levels of expression in any of the comparisons.

**Table 12 cancers-15-01420-t012:** Gene variants/mutations present in the cell lines investigated in this study.

		Chr	Base	CDS	AA		1°	Metastatic			
Gene	IGV ^‡^ Link	Location	Change	Change	Change	435	Tumor	Brain	Liver	Lung	Spine
DDX41	Chr 5: 176940410	5q35.3	T → C	1174 A → G	K392E	16.83	-----	-----	-----	-----	-----
GRIN2A	Chr 16: 9858034 ^†^	16p13.2	C → A	3367 G → T	E1123 *	26.97	-----	-----	-----	-----	-----
LILRB1	Chr 19: 55143437	19q13.42	C → A	410 C → A	S137 *	6.23	-----	-----	-----	-----	-----
PLCG1	Chr 20: 39794170	20q12	G → C	1590 G → C	E530D	12.85	-----	-----	-----	-----	-----
PCLO •	Chr 7: 82583661	7q21.11	AG → A	6607 del C	L2203 *	9.18	-----	-----	-----	-----	-----
NHS	Chr X: 17745083	Xq22.13	G → A	2794 G → A	D932N	-----	-----	-----	-----	6.62	-----
PIK3R1	Chr 5: 67593265	5q13.1	G → C	2011 G → C	V671L	-----	-----	-----	-----	5.70	-----
EIF4A1	Chr 17: 7478539	17p13.1	C → T	308 C → T	A103V	-----	-----	-----	11.51	-----	-----
EPHA2	Chr 1: 16455960	1q36.13	C → T	2794 G → A	A932T	-----	-----	-----	32.05	-----	12.60
ERCC3	Chr 2: 128046383	2q14.3	C → T	688 G → A	E230K	-----	-----	-----	31.8	-----	11.76
MKI67	Chr 10: 129905527	10q26.2	G → T	4577 C → A	A1526E	-----	31.21	33.06	-----	-----	17.95
PRKN	Chr 6: 162683593	6q26	C → T	376 G → A	D126N	-----	48.11	46.83	-----	-----	29.26
PCLO •	Chr 7: 82580341	7q21.11	G → A	9563 C → T	S3188F	-----	21.52	22.54	-----	-----	13.20
PCLO •	Chr 7: 82595727	7q21.11	G → A	3377 C → T	P1126L	-----	18.17	20.62	18.41	-----	19.05
POLE	Chr 12: 133225951	12q24.33	C → T	3946 G → A	G1316R	-----	47.56	46.92	48.44	-----	47.90
CDKN1C	Chr 11: 2905327	11q15.4	C → T	293 G → A	S98N	-----	31.48	34.38	30.84	-----	33.15
IGSF3	Chr 1: 117122200	1q13.1	C → A	3208 G → T	E1070 *	-----	54.31	53.63	53.28	45.95	52.67
MED12	Chr X: 70350053	Xq13.1	C → G	4036 C → G	R1346G	-----	44.67	45.56	50.27	43.53	49.22

^‡^ Abbreviations: IGV–Integrative Genomics Viewer; Chr–chromosome; CDS–coding sequence; AA–amino acid; 435–Parental 435 Cell Line; del–deletion. * Denotes non-sense. ^†^ Reference database ID = COSM1128836. • Denotes that three different changes were found in the PCLO gene in different cell lines.

## Data Availability

The datasets generated and analyzed during the current study are available from the corresponding authors upon request.

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
