# Peer review of "Isogenic Cell Lines Derived from Specific Organ Metastases Exhibit Divergent Cytogenomic Aberrations"

_cancers, 2023, doi:10.3390/cancers15051420_

Round 1

Reviewer 1 Report

Intratumor heterogeneity is one of the leading determinants of therapeutic resistance and treatment failure. Identification of common or organ specific changes may result in development of new diagnostic or therapeutic strategies. Most of studies on tumor heterogeneity focus on SNPs in DNA sequences or transcriptional profiles . This manuscript studied cytogenetic, biological processes, and SNPs, in combination with  previously characterized transcriptomic, proteomic, and metabolomic changes using the isogenic cell lines that were generated from a metastatic mouse model system. This may be one of the most comprehensive characterizations of tumor heterogeneity using a well-established model system. Most importantly, this manuscript not only identified some pan-metastatic therapeutic targets such as CNTOB and NCAPG2 and also demonstrated organ-specific aberration patterns. There are a few minor concerns as described below.

1.       The authors have identified many organ-specific aberrations. It would be interesting to discuss whether any similar changes have been reported in breast cancer metastasis of the corresponding organs.

2.       Another interesting finding is that higher numbers of chromosomes with SNP instability in the tumor, brain, and liver cell lines while, in general, fewer chromosomes with SNP instabilities were found in the lung and spine cell lines. It is not clear whether these patterns reflect the relative cellular proliferation rate of these metastatic cells or metastatic route?

Author Response

  1. The authors have identified many organ-specific aberrations. It would be interesting to discuss whether any similar changes have been reported in breast cancer metastasis of the corresponding organs.

Response:

We agree with the reviewer. Due to space constraints, it was not possible to go into detailed discussions of the consistencies of our findings with those found in clinical samplings studies. However, the issue has not been ignored, and the Reviewer will note that on page 18 we indicated that the bulk of the amplified, gained or loss genes have been reported within the context of breast cancer.   We also, pointed out that several reports concur with our S/G2-mitosis/biological processes results and have provided 11 citations on page 24 that can be referenced by those interested in more details on these findings.

  1. Another interesting finding is that higher numbers of chromosomes with SNP instability in the tumor, brain, and liver cell lines while, in general, fewer chromosomes with SNP instabilities were found in the lung and spine cell lines. It is not clear whether these patterns reflect the relative cellular proliferation rate of these metastatic cells or metastatic route?

Response:

In fact, from the biological processes/CIN data and SNP data one would expect to find compromised/delayed cell cycle/mitosis progression in the metastatic cell lines. We have done these experiments and the results have been reported; e.g., as calculated from the first 72 hrs of in vitro growth, the average length of the cell cycle division increased from 27 hrs for the 1o tumor cell line to 112, 38, 30, and 36 hrs for the brain, liver, lung, and spine cell lines respectively (see ref. 41).

Reviewer 2 Report

Authors report on cell lines derived from one single tumor and its metastasis and studied by different approaches

Comments
+ line 20-21: avoid the hint on model animal systems – the paper does not deal with animal but human model system
+ lines 89-90 – explain all abbreviations as DMEM, FBS, F12
+ line 105: authors need refer to newest version of ISCN – i.e. ISCN(2020) – and already since 2016 edition ISCN is no longer called “International System for Human Cytogenetic Nomenclature” but ‘Cytogenetic’ has been replaced by Cytogenomic. Please adapt.
+ acc. to ISCN there are no gaps allowed in a karyotype formula  - remove all gaps in all formulas given. Also (see line e.g. 226) – ‘and’ and ‘.’ are not allowed in a karyotype formula; one must write, +mar1,+mar2,+mar3 etc
+ overall state of the art is to perform besides GTG-banding also multicolor FISH analyses using all 24 whole chromosome paints as probes of such complex cell lines – this should be done, or if not possible it must be explained why this has been skipped. The interpretation on which chromosomes are lost and gained in how many copies is not possible in karyotypes with so many chromosomal markers without FISH.
+ Figure 4 is out of focus and the inscriptions M1, M4, M5, M7 and M3 disrupt one gray line; also it must read ‘add’ and not ‘Add’ in figure, which also includes “line end symbols” e.g. after add(1), which are not to be shown
+ Overall, the obtained proteomic data and SNP data is huge and hard to understand. It is more than clear that such complex karyotypic changes lead to massive changes in the metabolism of the affected cells. So what is the clue here? It is impossible in such a complex network to find ‘causative’ genes and very unlikely to find targetable genes. Especially it is not understandable why the genes which are impaired in similar ways all over all tumor subpopluations should be best suited targets of a therapy. The problem of solid tumor treatment is that subgroups with special adaptations escape treatment. So how can be excluded that any of the ‘not in common changes’ are not preadaptations of the tumor to potential treatments. Please revise the whole corresponding argumentation.

Reviewer 3 Report

Important, well-reasoned, and designed study, although I have several suggestions to consider:

Formatting inconsistencies, unnecessary gaps at the beginning of the sentences, from line 88 – different font size

Despite cell culture generation being described in detail in previous work, I would prefer having it briefly summarized also in methods of this article (for example original cell line used for the metastasis derivation), so I don’t have to read two papers instead of one.

Line 108 – the sentence is either too complicated for me or does not make sense, needs rewriting.

Line 102 – you mention that you compared fold change of expression between cell lines, however, no more information is gives as to what fold change and why did you choose. Fold change range is later mentioned on Results but it should be covered also in Methods.

Figure 1-3 is difficult to follow, I suggest labelling individual picture in figure by letters or number, so they are both easier to describe and follow.

Methods in general – number of samples/biological replicates sequenced or karyotyped should be mentioned. It can be partially determined from the results section, that you prepared five karyotypes from each type of tumor model, but it is unnecessary work for the reader and it would be easier to have it all summarized in Methods section. Moreover, it is unclear whether those five karyotypes come from five different animals or it is replicated specimen from one tumor. As both intra- and inter-tumoral heterogeneity is your main topic, I suggest making it clearer in the Methods.

Line 406 – potential – potentially

Figure 7A – Legend is illegible

Line 711 – Supplementary Materials statement is not filled

Round 2

Reviewer 2 Report

Thanks to authors for working on the paper.

Still following points need to be addressed:

- in simple summary authors still talk about an animal model.

- the use of a xenograph model is NOT described in MM part - please include.

- the argument that FISH is no longer used in the lab of the authors is not sufficient. Besides FISH can be done in cowork with other specialized labs, easily, or even can be get done by a comercial provider of such tests.
If authors used SNP-array they should also be able to include this data in their final karyotype interpretations, whch was not done yet. However, as SNP-array only detects imbalances and in human tumors balanced translocations are basic drivers of ongogene activation, they will not be able to characterize the cell line karyotypes comprehensively  this way. Authors seem to imply that FISH is outdated and could be completely replaced by NGS and or aCGH - this is not correct and cannot be correct as FISH has a single cell and single chromosome view and this is not available in the other approaches. Some recent papers in this MDPI journal showed nicely that cell line karoytpes can only be chatracterized by combining (molecular) cytogenetics and aCGH. Besides NGS might help in future, as soon as we learn how to deal with the flood of data available there.

So this point is still open - without FISH-data the cell line karyotypes cannot be interpreted.

- ISCN formulas are still not completely correct - ther are still some gaps too much, some commas lack, etc.

- The presentation of the overall data is still unclear and confusing, as are the conclusions, from point of view of this reviewer.

Round 3

Reviewer 2 Report

Paper is much better now

If editors accept lack of M-FISH - ok

Still  ISCN formulas are not completely correct - ther are still some gaps too much, some commas lack, etc.

This must be corrected

Author Response

Still  ISCN formulas are not completely correct - ther are still some gaps too much, some commas lack, etc.

Response:

Thanks for bringing this to our attention. We have corrected all the gaps as indicated earlier. Some of the issues are as result of typesetting. We did rectify it again in the revised submission.